# Active Observing in Continuous-time Control

**Samuel Holt**
University of Cambridge
sih31@cam.ac.uk

**Alihan Hüyük**
University of Cambridge
ah2075@cam.ac.uk

**Mihaela van der Schaar**
University of Cambridge
mv472@cam.ac.uk

## Abstract

The control of continuous-time environments while actively deciding when to take costly observations in time is a crucial yet unexplored problem, particularly relevant to real-world scenarios such as medicine, low-power systems, and resource management. Existing approaches either rely on continuous-time control methods that take regular, expensive observations in time or discrete-time control with costly observation methods, which are inapplicable to continuous-time settings due to the compounding discretization errors introduced by time discretization. In this work, we are the first to formalize the continuous-time control problem with costly observations. Our key theoretical contribution shows that observing at regular time intervals is not optimal in certain environments, while irregular observation policies yield higher expected utility. This perspective paves the way for the development of novel methods that can take irregular observations in continuous-time control with costly observations. We empirically validate our theoretical findings in various continuous-time environments, including a cancer simulation, by constructing a simple initial method to solve this new problem, with a heuristic threshold on the variance of reward rollouts in an offline continuous-time model-based model predictive control (MPC) planner. Although determining the optimal method remains an open problem, our work offers valuable insights and understanding of this unique problem, laying the foundation for future research in this area.

## 1 Introduction

The problem of continuous control with costly observations is ubiquitous with applications in medicine, biological systems, low power systems, robotics, resource management and surveillance [Yoshioka and Tsujimura, 2020, Brunereau et al., 2012, Mastronarde and van der Schaar, 2012]. A setting that is shared across all these domains is that a decision-maker needs to continually control (e.g., chemotherapy dosing, quantities of food, data to transmit, etc.) whilst deciding *when* to take a costly observation (e.g., a medical computed tomography (CT) scan, measuring the population of a species, measuring bandwidth, etc.). The decision-maker's observing policy must be timely to determine whether the action plan is effective (e.g., errors in treating stage 4 cancer can be fatal [Reinhardt et al., 2019], with further application examples in Appendix B).

In many of these real-world systems (e.g., medicine, low-power systems, resource management), an offline setup is beneficial as it enables decision-makers to learn control policies without incurring excessive costs or risking adverse effects. Where an offline setup refers to learning a policy from a previously collected dataset of state-action trajectories, without interacting with the environment [Argenson and Dulac-Arnold, 2020]. The development of novel methods that can take irregular observations in continuous-time control with costly observations, within an offline setup, provides a safe and cost-effective way to improve real-world decision-making processes.

Recent work falls into three main categories. First, *sensing approaches* determine when to informatively observe to identify an underlying state, but are unable to continually control. Second, *planning approaches* only continually control and have the restrictive assumption that the observing schedule is

*given*—most often at a regularly fixed observation time interval. While these methods can optimally solve many challenging tasks, with regular, frequent observations of the state, frequent observations are overly costly with many being unnecessary, and where the optimal regular observation interval is unknown.

Third, *discrete monitoring approaches* control and determine when to observe only for discrete time. These are fundamentally incompatible with continuous-time systems which can exhibit irregular observations in time and states that evolve on different time scales. Moreover, these rely on a discretization interval being carefully selected to use but still suffer from time discretization errors that can compound, leading to unstable and poor-performing controllers [Yildiz et al., 2021]. For example, in medicine, a crucial test may be administered too late, and therefore delay updating the treatment dosing—which can be fatal when treating patients for cancer [Geng et al., 2017].

**Contributions:** ① We are the first to formalize the problem of continuous-time control whilst deciding when to take costly observations. Theoretically, we show that regular observing in continuous time with costly observations is not optimal for some systems and that irregularly observing can achieve a higher expected utility (Section 2). ② Empirically we verify this key theoretical result in a cancer simulation and standard continuous-time control environments with costly observations. We construct a simple initial method to solve this new problem, called Active Observing Control. This uses a heuristic threshold on the variance of reward rollouts in an offline continuous-time model-based model predictive control (MPC) planner (Sections 4 and 5.1). However, the optimal method is still an open problem, and we leave this for future work. We gain insight into this unique problem, uncovering how our initial method is capable of irregularly observing, and identifies observation frequencies that correlate to different stages of cancer fixed observing frequencies of clinicians. Further, we demonstrate how it can avoid discretization errors in time, thus achieving a better utility as a result (Section 5.2), and confirm its robustness to its threshold hyperparameter (Section 5.3).

We have released a PyTorch [Paszke et al., 2019b] implementation of the code at https://github.com/samholt/ActiveObservingInContinuous-timeControl and have a broader research group codebase at https://github.com/vanderschaarlab/ActiveObservingInContinuous-timeControl.

## 2   Problem

**States & actions.** For a system with *state* space $\mathcal{S} = \mathbb{R}^{d_{\mathcal{S}}}$ and *action* space $\mathcal{A} = \mathbb{R}^{d_{\mathcal{A}}}$, the state at time $t \in \mathbb{R}_+$ is denoted as $s(t) = [s_1(t), \ldots, s_{d_{\mathcal{S}}}(t)] \in \mathcal{S}$ and the action at time $t \in \mathbb{R}$ is denoted as $a(t) = [a_1(t), \ldots, a_{d_{\mathcal{A}}}(t)] \in \mathcal{A}$. We elaborate that a *state trajectory* $s \in \mathcal{S}^{\mathbb{R}}$ and an *action trajectory* $a \in \mathcal{A}^{\mathbb{R}}$ are both functions of time, where an individual state $s(t) \in \mathcal{S}$ or an individual action $a(t) \in \mathcal{A}$ are points on these trajectories. In practical applications, action values are usually bounded by an actuator's limits; hence we also restrict the action space to a Euclidean box $\mathcal{A} = [a_{\min}, a_{\max}]^{d_{\mathcal{A}}}$.

**Environment dynamics.** The dynamics of the system are given by $ds(t)/dt = f(s(t), a(t))$, and the system is stationary. We consider the setting where the true state trajectory $s$ is latent and, at a given time $t$, only a noisy observation $z(t) = s(t) + \varepsilon(t) \in \mathcal{S}$ can be observed, where $\varepsilon(t) \sim \mathcal{N}(0, \sigma_\epsilon^2)$ is Gaussian noise with zero mean and standard deviation $\sigma_\epsilon$ and $\varepsilon(t) \perp\!\!\!\perp \varepsilon(t'), \forall t, t' \in \mathbb{R}_+$. We denote as a tuple $(t, z(t), a(t))$ a discrete sample taken at time $t$, and with $h = \{(t_j, z(t_j), a(t_j))\} \in \mathcal{H}$ a history of samples taken at times $\{t_j\}$ where $\mathcal{H} = \cup_{n=0}^{\infty} (\mathbb{R}_+ \times \mathcal{S} \times \mathcal{A})^n$ is the space of all possible histories.

---

**Algorithm 1** Policies $\rho, \pi$ interacting with the environment

1: $t_1 = 0, h_1 = \{(t_1, z(t_1), a(t_1))\}$
2: **for** $i \in \{1, 2, \ldots\}$ **:**
3:    Schedule next observation:    $t_{i+1} = t_i + \rho(h_i)$
4:    Execute actions: $a(t) = \pi(h_i, t - t_i)$ for $t \in [t_i, t_{i+1})$
5:    Take an observation:   $h_{i+1} = h_i \cup \{(t_{i+1}, z(t_{i+1}), a(t_{i+1}))\}$

---

**Policies.** Policies consist of an *observing policy* $\rho : \mathcal{H} \to \mathbb{R}$ and a *decision policy* $\pi : \mathcal{H} \times \mathbb{R} \to \mathcal{A}$. These policies interact with the environment as described in Algorithm 1. We denote with $\Delta_{j+1} = t_{j+1} - t_j$, where $j$ is a dummy variable.

**Objective.** Suppose we are given a reward function $r : \mathcal{S} \to \mathbb{R}$ to maximize, and each observation has a fixed cost $c \in \mathbb{R}_+$. Then, the utility achieved by a policy up to a time horizon $T$ is given by

$$\mathcal{U} = \underbrace{\int_0^T r(s(t), a(t), t) \mathrm{d}t}_{\text{Reward } \mathcal{R}} - \underbrace{c|\{t_i : t_i \in [0, T]\}|}_{\text{Cost } \mathcal{C}} \tag{1}$$

Our objective is to find the optimal observing and decision policies $\rho^*, \pi^* = \arg\max_{\rho, \pi} \mathbb{E}[\mathcal{U}]$ given an offline dataset of histories $\mathcal{D} = \{h^{(i)}\}$ but without knowing $f$ and $\sigma_\epsilon$ or having online access to the system described by $f$ and $\sigma_\epsilon$.

## 2.1 Taking regular observations in time is not optimal for some systems

This is a highly non-trivial task since, in many cases, the obvious solution of taking regular observations in time is not optimal, and rather taking irregular observations achieves a higher expected utility—a point we make formally in Proposition 2.1.

**Proposition 2.1.** *For some systems, it is not optimal to observe regularly—that is $\exists f, \sigma_\epsilon, r, c, h, h'$ : $\rho^*(h) \neq \rho^*(h')$.*

*Proof.* Full proof is in Appendix C. However, we present the following sketch. Consider the task of maximizing the utility $\mathcal{U}$ given a reward function $r(s, a, t) = \delta(t - T) \cdot \mathbb{1}\{s(t) \cdot a(t) > 0\}$, observation cost $c > 0$ for the specific system $ds/dt = s(t)$ with observation noise ($\sigma_\epsilon > 0$). Intuitively this reward function only provides a positive reward of $+1$ at the end of the episode ($t = T$) if the chosen action $a(T)$ has the same sign as the unobserved state $s(T)$, otherwise, the reward is $0$. Therefore, this $r$ is maximized when the sign of the correct state $s(t)$ is identified. Since this system is defined by $ds/dt = s(t)$, we know that state solutions are of the form $s(t) = s(0)e^t$. Thus, to determine the sign of $s(t)$, since $e^t$ is always strictly positive, the sign of the state is determined by the initial condition $s(0)$. Our observing policy can observe anywhere inside the interval of $[0, T]$. We prove that the optimal observing policy, $\rho^*$ cannot be a regular observing policy by showing that, for each $\rho^{(\text{reg}, \delta)}$, there exists at least one $\rho^{(\text{irreg}, \ell)}$ with $\ell \geq 2$ that achieves higher expected utility, where $\ell$ is the number of observations taken. A higher expected utility can be achieved by an irregular observing policy that takes all observations at the end of the episode, i.e. $\rho^*(h_i) = T - t_i$ for $i \in \{2, \ldots, \ell\}$—that is when the signal-to-noise ratio ($s(t)^2/\sigma_\epsilon^2$) is the highest, i.e. $\arg\max_t \frac{s(0)^2 e^{2t}}{\sigma_\epsilon^2} : t \in [0, T]$, which occurs when $t = T$. $\qquad\square$

This motivates us to develop new continuous-time methods for solving this problem, where a solution should be able to flexibly adapt to take irregular observations when it achieves a higher expected utility. Intuitively, based on the above system, it is more informative to take an observation when the predicted state change magnitude is larger, which can occur faster for a large state velocity $\dot{s}$—indicating observations should be taken more frequently. We later show experimentally illustrative cases (Section 5.2) of how irregularly observing can lead to better performance.

## 3 Related work

Table 1 summarizes the key differences between Active Observing Control from related approaches to planning and monitoring in reinforcement learning (RL). Moreover, we provide an extended related work in Appendix D, which includes discussions of the benefits of offline RL, model-based RL, why model predictive control is our preferred action policy, event-based sampling, Semi-MDP methods, and Linear Quadratic Regression (LQR) & Linear Quadratic Gaussian (LQG) methods.

**Sensing** approaches have been proposed of when to optimally take an observation in both discrete time [Jarrett and Van Der Schaar, 2020] and continuous time [Alaa and Van Der Schaar, 2016, Barfoot et al., 2014]—where their goal is to identify an underlying state. However, these approaches cannot also continuously control the system. In contrast, Active Observing Control seeks to both actively observe and control the system.

**Planning** approaches only focus on optimal decision policies $\pi$, and therefore observing has to be provided by a schedule, which is often assumed to be at a regular constant interval in time $\Delta_{i+1} =$

Table 1: **Comparison with related approaches to planning and monitoring in RL.** Plots for the corresponding state trajectories for each approach are illustrated in Figure 1. Our initial method, Active Observing Control, is the only method for continuous-time control whilst deciding *when to observe* with *smoothly-varying* states.

| Approach | Ref. | Domain | Environment Dynamics | State Trajectories | Policies | Policy Formulation |
|---|---|---|---|---|---|---|
| Discrete Sensing | [Jarrett and Van Der Schaar, 2020] | Discrete-time | $s_i \sim s_{\text{constant}}$ | A | Observing-only | $i+1 = i + \rho(h_i)$ |
| Discrete Planning | [Chua et al., 2018] | Discrete-time | $s_{i+1} \sim f(s_i, a_i)$ | B | Decision-only | $a_i = \pi(h_i)$ |
| Discrete Monitoring | [Nam et al., 2021] | Discrete-time | $s_{i+1} \sim f(s_i, a_i)$ | C | Decision & observing | $a_{i' \in \{i,\dots,i+\rho(h_i)\}} = \pi_{i'-i}(h_i)$ |
| Continuous Sensing | [Alaa and Van Der Schaar, 2016] | Continuous-time | $s(t) \sim s_{\text{constant}}$ | D | Observing-only | $t_{i+1} = t_i + \rho(h_i)$ |
| Continuous Planning | [Yildiz et al., 2021] | Continuous-time | $\mathrm{d}s(t)/\mathrm{d}t \sim f(s(t), a(t))$ | E | Decision-only (w/ regular observations) | $a(t \in [t_i, t_i + \Delta]) = \pi(h_i, t - t_i)$ |
| Semi-continuous Monitoring | [Huang et al., 2019] | Continuous-time | $s(t \in [t_k, t_{k+1}]) \sim s_k$ $(s_{k+1}, t_{k+1} - t_k) \sim \tilde{f}(s_k, a(t \in [t_k, t_{k+1}]))$ | F | Decision & observing | $a(t \in [t_i, t_i + \rho(h_i)]) = \pi(h_i, t - t_i)$ |
| **Active Observing Control** | **(Ours)** | Continuous-time | $\mathrm{d}s(t)/\mathrm{d}t \sim f(s(t), a(t))$ | G | Decision & observing | $a(t \in [t_i, t_i + \rho(h_i)]) = \pi(h_i, t - t_i)$ |

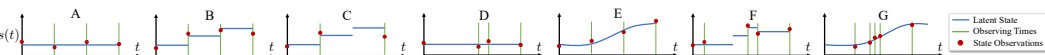

Figure 1: Comparison of related approaches of an environment's latent state $s(t)$—where green lines represent observing times and red dots observations that have observation noise.

$\Delta_i$—that is, observations are not actively planned [Yildiz et al., 2021]. Within planning approaches, there exist many discrete-time approaches [Chua et al., 2018, Mnih et al., 2013, Williams et al., 2017] and recently more continuous-time approaches [Du et al., 2020, Yildiz et al., 2021]. Specifically, Yildiz et al. [2021] presented a seminal online continuous-time model-based RL algorithm, leveraging a continuous-time dynamics model that can predict the next state at an arbitrary time $s(t)$. However, all these methods are *unable* to plan when to take the next observation, in contrast to Active Observing Control which *is* able to determine when to take the next observation whilst planning an action trajectory.

**Monitoring** approaches consist of both a decision policy $\pi$ and an observing policy $\rho$; however, existing approaches only consider the discrete-time and discrete-state setting with observation costs [Sharma et al., 2017, Nam et al., 2021, Aguiar et al., 2020, Bellinger et al., 2020a, Krueger et al., 2020, Bellinger et al., 2020b]. In particular, Krueger et al. [2020] proposed that even in a simple setting of a discrete-state multi-armed bandit, computing the optimal time to take the next observation is intractable—therefore, they must rely on heuristics for their observing policy. Broadly, discrete-time monitoring methods use a discrete-time model and propagate the last observed state, often appending either a flag or a counter to the state-action tuple to indicate the time since an observation was taken [Aguiar et al., 2020, Bellinger et al., 2020b]. However, these methods cannot be applied to continuous-time environments or propagate the predicted current state and its associated prediction interval of the uncertainty associated with the state estimate. Moreover, training a policy [Aguiar et al., 2020] that decides at each state whether to observe it or not, requires a discretization of time. Whereas Active Observing Control, which determines the continuous-time interval of when to observe the next state, does not require any discretization of time, and hence it is a continuous-time method and avoids compounding time discretization errors.

One approach exists that we term *semi-continuous monitoring*, where Huang et al. [2019] proposes a discrete-state, constant action policy, that determines when to take an observation in continuous time of a controlled Markov Jump Process. However, this approach is limiting, as it assumes actions are constant until the next sample of the state is taken [Ni and Jang, 2022]—which is clearly suboptimal, uses discrete states, and assumes a Markov Jump Process [Gillespie, 1991]. Instead, Active Observing Control is fully continuous in both states and observing times—controlled by an action trajectory $a$, giving rise to *smoothly-varying* states.

## 4 Active Observing Control

We now propose Active Observing Control (AOC), an initial method for the defined problem of continuous-time control whilst deciding when to take costly observations. AOC, can plan action trajectories in continuous time and plan when to observe next in continuous time. The key idea is to use a *continuous-time uncertainty-aware* dynamics model $\hat{f}_\theta$ to plan 1) action trajectories and 2) the next time to take an observation, such that it is informative to do so—Figure 2 provides a block diagram. We adhere to the standard offline model-based setup [Lutter et al., 2021] of first learning our dynamics model (Section 4.1), and then using it at run-time with our planning method (Section 4.2).

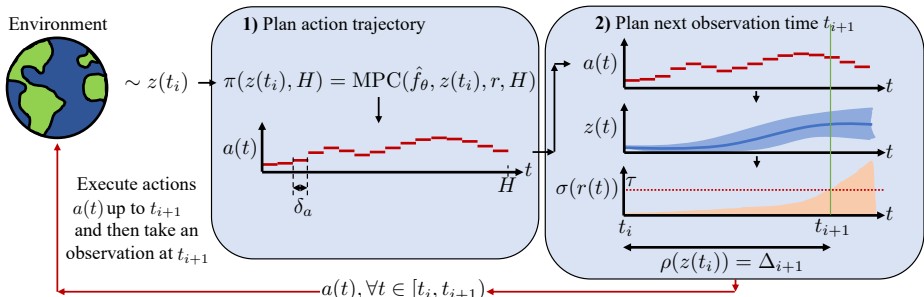

Figure 2: **Block diagram of Active Observing Control.** An uncertainty-aware dynamics model $\hat{f}_\theta$ is learned from an offline dataset $\mathcal{D}$ of state-action trajectories. At run-time, planning consists of two steps: **1)** The actions are determined by a Model Predictive Control (MPC) Planner, and **2)** the determined action trajectory $a$ is forward simulated to provide uncertainty $\sigma(r(t))$ on the planned path reward. We determine the continuous time $t_{i+1}$ to execute the action plan $a$ up to, such that $\sigma(r(t)) < \tau$, as in Algorithm 3. We then execute $a(t) \ \forall t \in [t_i, t_{i+1})$ up until the time to take the next observation $z(t)$ at.

## 4.1 Learning a continuous-time uncertainty-aware dynamics model

Fundamentally the goal of offline model-based RL is to learn an *accurate* dynamics model that can be used for planning from an offline dataset $\mathcal{D}$ [Lutter et al., 2021]. In particular, we require a dynamics model that is both uncertainty aware and continuous in time; that is, it can provide a prediction uncertainty for the next state at a future time $t + \delta$, i.e. $p(z(t + \delta)|(z(t), a(t), \delta))$. Here we use the time difference $\delta$ input to create a continuous-time dynamics model [Yildiz et al., 2021].

Model-based RL has shown the crucial importance of the performance of learning an uncertainty-aware dynamics model that captures both 1) *aleatoric* uncertainty—due to the inherent stochasticity of the environment (e.g., observation noise and environment process noise) that is irreducible and 2) *epistemic* uncertainty—due to the lack of data, for a given state-action space (which should vanish when trained on the limit of infinite data) [Chua et al., 2018].

We learn an uncertainty-aware dynamics model by training an ensemble of high-capacity multi-layer perceptron (MLP) neural network models that each parameterize a multivariate Gaussian distribution—where the ensembling captures the *epistemic* uncertainty and each model individually captures the *aleatoric* uncertainty. We note that as shown by other works, ensembles of high-capacity neural networks outperform Gaussian process dynamics models, as they have constant-time inference—scaling better to larger offline datasets, while still providing well-calibrated uncertainty estimates [Lakshminarayanan et al., 2017] and can model more complex functions—inclusively non-smooth dynamics [Chua et al., 2018].

Precisely, the ensemble consists of $M = 5$ neural network models that output the parameters for a multivariate Gaussian distribution with diagonal covariances, each with $\theta_m$ parameters, i.e., $\hat{f}_{\theta_m} = p_{\theta_m}(z(t+\delta)|(z(t), a(t), \delta)) = \mathcal{N}(\mu_{\theta_m}(z(t), a(t), \delta), \Sigma_{\theta_m}(z(t), a(t), \delta))$ where the elements of the diagonal covariances are given by $\sigma^2_{\theta_m}(z(t), a(t), \delta)$, with a total of $\theta = \{\theta_m\}_{m=1}^M$ parameters of the ensemble. Moreover, we denote all the models in the ensemble as $\hat{f}_\theta$.

To create our ensemble of parameterized Gaussian distribution models, we train each model independently with unique initialized random weights on a unique permutation of all the samples in the offline dataset—whereby each base model converges to its own local optima and is more effective than training the models on subsets of the dataset, i.e., bootstrapping [Lakshminarayanan et al., 2017]. Therefore, we minimize the negative log-likelihood for every single model separately, that is

$$\mathcal{L}(\theta_m) = \sum_{h \in \mathcal{D}} \sum_{j=1}^{|h|-1} [\mu_{\theta_m}(z(t_j), a(t_j), \delta_{j+1}) - z(t_{j+1})]^\top \Sigma_{\theta_m}^{-1}(z(t_j), a(t_j), \delta_{j+1})$$

$$[\mu_{\theta_m}(z(t_j), a(t_j), \delta_{j+1}) - z(t_{j+1})] + \log \det \Sigma_{\theta_m}(z(t_j), a(t_j), \delta_{j+1}),$$

where $\delta_{j+1} = \Delta_{j+1} = t_{j+1} - t_j$ arises from the offline dataset $\mathcal{D}$. Here each parameterized Gaussian distribution model captures heteroskedastic (i.e., the output noise distribution is a function of the

input) *aleatoric* uncertainty. To capture the heteroskedastic *epistemic* uncertainty, we combine the individual models as a uniformly weighted mixture model and combine the predictions as $p_\theta(z(t + \delta)|(z(t), a(t), \delta), \theta) = \frac{1}{M} \sum_{m=1}^{M} p_{\theta_m}(z(t + \delta)|(z(t), a(t), \delta), \theta_m)$. Therefore, we can compute the effective mean and covariance (diagonal elements) of the mixture as

$$\mu_*(z(t), a(t), \delta) = \mathbb{E}_m[\mu_{\theta_m}(z(t), a(t), \delta)]$$
$$\sigma_*^2(z(t), a(t), \delta) = \mathbb{E}_m[(\sigma_{\theta_m}^2(z(t), a(t), \delta) + \mu_{\theta_m}^2(z(t), a(t), \delta)] - \mu_*^2(z(t), a(t), \delta)$$

## 4.2 Active Observing Control Planning

We desire to use the trained uncertainty-aware dynamics model $\hat{f}_\theta$ to **1)** plan the optimal action trajectory $a$ to execute and **2)** plan when best to take the next observation. Intuitively, we use the probabilistic dynamics model $\hat{f}_\theta$ to plan an optimal action trajectory $a$ and execute this until the next observing time—as determined by the heuristic of when the predicted reward distribution over the planned trajectory crosses a set threshold, Figure 2. In the following, we detail these two steps.

**1) Planning optimal actions.** To plan an optimal action trajectory $a$, we specify that any model predictive controller (MPC) planner can be used—that can optimize the action trajectory up to a specified time horizon $H \in \mathbb{R}_+$. We opt to use an MPC planner, as it optimizes an entire action trajectory $a$ up to $H$, does not require the probabilistic dynamics model $\hat{f}_\theta$ to be differentiable (it uses no gradients), and the reward function $r$ can be changed on the fly—allowing changing goals/tasks at run-time. We use our probabilistic dynamics model $\hat{f}_\theta$, with the MPC planner of Model Predictive Path Integral Control (MPPI), a zeroth-order particle-based trajectory optimizer [Williams et al., 2017], due to its competitive performance [Wang et al., 2022].

To optimize the action trajectory $a$ up to a time horizon $H$, it discretizes $H$ into smaller constant action time intervals $\delta_a \in \mathbb{R}_+$ which are then optimized—where there are $K = \lceil \frac{H}{\delta_a} \rceil \in \mathbb{Z}_+$ time steps in $H$—i.e., $t^{(k)} = t_i + k\delta_a, k \in \{0, \ldots, K - 1\}$. It forward simulates several parallel rollouts $G \in \mathbb{Z}_+$, where the next state estimate at each time step is simulated as $z(t^{(k+1)}) = \mu_*(z(t^{(k)}), a(t^{(k)}), \delta_a)$ recursively. This requires a state estimate $z(t_i)$ to plan from. Therefore, we recompute the planned trajectory when we observe a new sample of the state. Furthermore, we provide MPC MPPI planner pseudocode and details in Appendix E.

**2) When to observe.** We desire to observe when it is most informative to do so. Therefore, we seek to determine the time interval $\Delta_{i+1}$ that the action trajectory $a$ can be followed until an observation is taken. Intuitively, we create a reward distribution over continuous time following the MPC planned action trajectory $a$, which we can follow until the uncertainty crosses a set threshold $\tau \in \mathbb{R}_+$—a hyperparameter to be tuned. Intuitively we prefer the reward uncertainty rather than that of the state because we can achieve the task with a very certain reward despite having uncertain states. For instance, when there might be multiple ways to achieve a task—and we know that our multiple action trajectories guarantee this where we can take any; however, are uncertain about which one. We empirically verify this in Appendix L.1.

To create the continuous-time reward distribution and state distribution, we use our learned probabilistic dynamics model $\hat{f}_\theta$ to forward propagate the last observed state, according to the planned action trajectory $a$. As noted by others [Chua et al., 2018], computing a closed-form expression for the distribution of the expected state trajectory is generally intractable; therefore, it is common to approximate the uncertainty propagation [Chua et al., 2018, Girard et al., 2002, Candela et al., 2003]. We generate this through Monte Carlo sampling by forward simulating $P \in \mathbb{Z}_+$ particle rollouts of state trajectories. That is, for a given state particle, for the next step of $\delta_a$ we sample a new state particle $z_p(t^{(k+1)}) \sim \mathcal{N}(\mu_*(z_p(t^{(k)}), a(t^{(k)}), \delta_a), \sigma_*^2(z_p(t^{(k)}), a(t^{(k)}), \delta_a))$ recursively along the planned action trajectory. This allows us to apply the reward function to the state particles, generating reward particles—whereby we can compute the mean and standard deviation of the reward at each time step $t^{(k)}$, i.e., $r(t^{(k)}) \sim \mathcal{N}(\mu_r(t^{(k)}), \sigma_r^2(t^{(k)}))$. Therefore, using our estimate of $\sigma(r(t^{(k)}))$ over the state trajectory $a$ allows us to determine the maximum time interval $\Delta_{i+1}$ until the reward uncertainty becomes greater than a set threshold $\tau$. Thus, the next time to observe at is

$$\rho(z(t_i)) = \max\{\Delta' \in \mathbb{R}_+ : \sqrt{\mathbb{V}_{z_p}[r(t_i + \Delta')]} < \tau\}$$

This provides an estimate of the next observation interval $\Delta_{i+1}$ that is discretized to an integer number of $\delta_a$ intervals. However, we seek the continuous-time observation interval $\Delta_{i+1}$; therefore,

we further refine this estimate through a continuous binary search (root finding algorithm) up to a tolerance of $\delta_t \in \mathbb{R}_+$. We outline the AOC planning algorithm pseudocode in Appendix F.

**Run-time complexity.** Intuitively, we find if the time taken to plan using an MPC controller is feasible in a problem setting, then AOC is feasible. As the run-time complexity of AOC is $O(GK + P(K + W))$ where $W = \lceil (\log(\delta_a) - \log(\delta_t))/\log(2) \rceil \in \mathbb{Z}_+$ (Appendix G). Empirically we chose $P = 10G, W = 4, G = 10,000$ for all experiments. Although AOC takes approximately $2.4\times$ longer to plan, which includes both planning the actions and when to observe compared to Continuous Planning methods that only plan actions (an MPC step)—it can often take fewer observations overall, leading to less time spent planning overall. AOC is still a practical method that can plan in less time than the action interval $\delta_a$, even in robotics environments, where it takes 24.9 ms to plan the actions and the next observation time (Appendix G).

An important parameter is the uncertainty threshold $\tau$ hyperparameter. We find the following procedure to tune this parameter for a benchmark sufficient. For a particular environment after training its dynamics model, one episode is run, where the action trajectory $a$ is re-planned at every $\delta_a$ time step and compute the median of the reward uncertainty over time for each rollout action plan, and then takes the mean of these over an episode to produce $\tau$. This step is akin to a calibration step, where calibration is common in medicine [Preston et al., 1988] and robotics [Nubiola and Bonev, 2013]. We also include practical guidelines to select $c$ in Appendix H.

# 5 Experiments and Evaluation

**Benchmark environments**. We use four continuous-time control environments and adapt them to add a fixed cost $c$ for taking an observation—these environments were selected as they exhibit a range of dynamical system state region regimes. First, the Cancer environment uses a simulation of a Pharmacokinetic-Pharmacodynamic (PK-PD) model of lung cancer tumor growth [Geng et al., 2017] under continuous dosing treatment of chemotherapy and radiotherapy. We note that the same underlying model has also been used by others [Lim, 2018, Bica et al., 2020, Seedat et al., 2022]. Moreover, we use the standard continuous-time control environments from the ODE-RL suite [Yildiz et al., 2021], which consists of three environments: Pendulum, CartPole, and Acrobot. We note that all the environments are described by a differential equation (DE) and use a DE solver to simulate the fully continuous in-time states and actions, unlike discrete environments [Yildiz et al., 2021, Brockman et al., 2016]. Furthermore, to increase realism, we add Gaussian noise to observations taken, that of $\mathcal{N}(0, \sigma_\epsilon^2)$ with standard deviation $\sigma = 0.01$. Specifically, the ODE-RL starting states have the poles hanging down at the stable equilibrium of the DE system, and the goal is to swing up and stabilize the pole(s) upright to the unstable equilibrium [Yildiz et al., 2021]. Additionally, the reward function form is the same across the environments—i.e., the exponential of the negative distance from the current state to the desired goal state $s^*$, while penalizing the action magnitudes— and we assume we are given this simple reward function when planning (however, we also show AOC can work with a learned reward model in Appendix L.9). Furthermore, we generate an offline dataset $\mathcal{D}$ from these environments that has irregular times between state-action samples $\Delta_{i+1} \sim \text{Exp}(\bar{\Delta})$, with a mean of $\bar{\Delta} = \delta_a$ seconds. We detail all environments, including offline dataset generation in Appendix I.

**Benchmark methods**. We seek to provide more competitive benchmarks than those outlined in Table 1; therefore, we compare against the following ablations of our method, **Active Observing Control**. Specifically, we focus on benchmarking observing policies $\rho$ and use the same action policy $\pi$ across all the benchmark methods, that of the same MPC MPPI planner with the hyperparameters fixed. We implement two

Table 2: Benchmark observing policies

| Benchmark | Observing Policy |
|---|---|
| Discrete Planning | $\rho(z(t_i)) = \delta_a$ |
| Discrete Monitoring | $\rho(z(t_i)) = \delta_a \max\{k : \sigma(t_i + k\delta_a) < \tau\}$ |
| Continuous Planning | $\rho(z(t_i)) = \delta_a$ |
| **Active Observing Control** | $\rho(z(t_i)) = \max\{\Delta : \sigma(t_i + \Delta) < \tau\}$ |

discrete-time methods by first learning a discrete-time uncertainty-aware dynamics model (an ablation of the same model and hyperparameters, without the time interval input $\delta$ to predict the next state for) on the same offline dataset $\mathcal{D}$. Second, we use this discrete-time uncertainty-aware dynamics model to create two benchmark methods, that of **Discrete Planning** that observes at each action time interval $\delta_a$ and **Discrete Monitoring** a discrete ablation of our observing policy (Algorithm 3)—that uses the same reward distribution estimation and determines the discrete action sequence to execute up until the reward uncertainty crosses the threshold $\tau$—at a discrete-time resolution of $\delta_a$. Moreover,

Table 3: Normalized utilities $\mathcal{U}$, rewards $\mathcal{R}$ and observations $\mathcal{O}$ for the benchmark methods, across each environment. AOC performs the best across all environments. Results are averaged over 1,000 random seeds, with $\pm$ indicating 95% confidence intervals. Utilities and rewards are undiscounted and normalized to be between 0 and 100, where 0 corresponds to a Random agent, and 100 corresponds to the expert, that of Continuous Planning, taking state observations at every $\delta_a$.

| Policy | Cancer | | | Acrobot | | | Cartpole | | | Pendulum | | |
| | $\mathcal{U}$ | $\mathcal{R}$ | $\mathcal{O}$ | $\mathcal{U}$ | $\mathcal{R}$ | $\mathcal{O}$ | $\mathcal{U}$ | $\mathcal{R}$ | $\mathcal{O}$ | $\mathcal{U}$ | $\mathcal{R}$ | $\mathcal{O}$ |
|---|---|---|---|---|---|---|---|---|---|---|---|---|
| Random | 0±0 | 0±0 | 13±0 | 0±0 | 0±0 | 50±0 | 0±0 | 0±0 | 50±0 | 0±0 | 0±0 | 50±0 |
| Discrete Planning | 91.7±0.368 | 91.7±0.368 | 13±0 | 87.1±1.05 | 87.1±1.05 | 50±0 | 83.6±0.56 | 83.6±0.56 | 50±0 | 87.2±0.962 | 87.2±0.962 | 50±0 |
| Discrete Monitoring | 91±0.532 | 85.8±0.522 | 5.08±0.0327 | 89.6±1.02 | 80.2±1.14 | 43.7±0.189 | 127±0.846 | 82.9±0.532 | 42.3±0.107 | 130±2.52 | 87.3±0.957 | 42.1±0.293 |
| Continuous Planning | 100±0.153 | 100±0.153 | 13±0 | 100±0.462 | 100±0.462 | 50±0 | 100±0.772 | 100±0.772 | 50±0 | 100±0.904 | 100±0.904 | 50±0 |
| **Active Observing Control** | **105±0.18** | 98.8±0.169 | 3.37±0.0302 | **107±0.911** | 90.8±0.878 | 39±0.177 | **151±1.54** | 99.5±0.774 | 41.1±0.196 | **177±2.18** | 98.8±0.912 | 35.6±0.239 |

we also benchmark against **Continuous Planning** that uses our trained continuous-time uncertainty-aware dynamics model and takes observations at regular time intervals of a multiple of $\delta_a$. Finally, we also compare with a random action policy, **Random** that observes at each action interval $\delta_a$. For ease of comparison, we list these observing policies in Table 2. We further provide the dynamics model implementation details, hyperparameters, and training details in Appendix J.

**Evaluation.** An uncertainty-aware dynamics model is trained for each environment on an offline dataset $\mathcal{D}$ of collected trajectories consisting of $10^6$ samples. The trained dynamics model weights are then frozen when used at run-time. We record the average undiscounted utility $\mathcal{U}$ (Equation (1)), average undiscounted reward $\mathcal{R}$ and the number of observations taken $\mathcal{O}$, after running one episode of the benchmark method—and repeat this for 100 random seed runs for each result with their 95% confidence intervals throughout. Moreover, we normalize $\mathcal{R}$ and $\mathcal{U}$ following the standard normalization of offline-RL [Yu et al., 2020]—normalized to the interval of 0 to 100, where a score of 0 corresponds to a random policy performance, and 100 to an expert—which we assume here is the Continuous Planning benchmark. Furthermore, we detail these metrics and the experimental setup further in Appendix K.

## 5.1 Main results

We evaluated all the benchmark methods across all our environments with results tabulated in Table 3. Active Observing Control achieves high average utility $\mathcal{U}$ on all environments. Specifically, AOC can achieve near-optimal state control performance $\mathcal{R}$ while using fewer observations $\mathcal{O}$ compared to taking regular, frequent observations of Continuous Planning. Furthermore, it outperforms Discrete

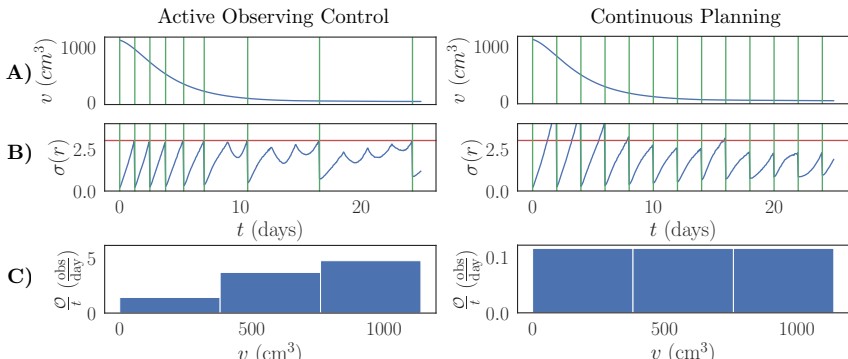

Figure 3: **Comparison of Active Observing Control against Continuous Planning on the cancer environment for one episode.** Rows: A) Cancer volume $v$ state trajectory—with green vertical lines representing observation times. B) Reward uncertainty of the planned state trajectory $a$ after an observation is taken—where the red horizontal line indicates AOC's threshold used $\tau$. C) Bar chart of the frequency of observations per state region. AOC automatically determines to observe larger cancer volumes more frequently as they are more informative, as the future state change magnitude is larger (Section 2.1)—which correlates to clinician's findings [Geng et al., 2017]. Whereas with Continuous Planning the observing frequency is regular, and therefore observations can be taken when the reward is still certain, suggesting an unnecessary observation.

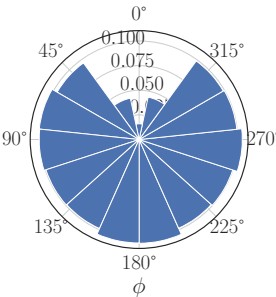

Figure 4: **Frequency of observations per state region for Pendulum.** The intuition from Section 2.1 indicates that it is more informative to sample when $\dot{\phi}$ is larger, hence near the top goal equilibrium point $\dot{\phi} \approx 0$ necessitates infrequent observations.

Table 4: Normalized utilities $\mathcal{U}$ and rewards $\mathcal{R}$ for the cancer environment—using the same normalization as in Table 3. Even when Continuous Planning takes the same number of observations as determined by AOC, it still performs worse, because those observations are not *well located*. As Continuous Planning observations are taken blindly at regular times, rather than at more informative points.

| Policy | Cancer | | |
| --- | --- | --- | --- |
| | $\mathcal{U}$ | $\mathcal{R}$ | $\mathcal{O}$ |
| **Active Observing Control** | 105±0.183 | 98.8±0.173 | 3.39±0.0306 |
| Continuous Planning with $\mathcal{O} = 3$ | 102±0.234 | 95.6±0.234 | 3±0 |
| Continuous Planning with $\mathcal{O} = 4$ | 103±0.226 | 97.3±0.226 | 4±0 |

Monitoring as it can determine when, in continuous time to observe, whereas those methods suffer from discretization errors—that compound, leading to worse performance.

## 5.2 Insight experiments

In the following, we gain insight into why AOC performs better than Continuous Planning, i.e., better than taking regular observations in time and the importance of continuous-time monitoring as opposed to discrete-time monitoring.

**How does irregularly observing achieve a higher expected utility than regularly observing?** To better understand why monitoring (active observing) approaches outperform planning that regularly observes, we provide a qualitative and quantitative study.

On the one hand, qualitatively, as detailed in Figure 3, we observe that for a given threshold, AOC can still solve the Cancer environment while taking less costly observations and that AOC automatically determines to increase the observing frequency when the tumor volume $v$ is large and less when it is smaller. Crucially, this matches what clinicians have already discovered, where they categorize the cancer volume into five discrete bins where the observing frequency is correlated to tumor volume [Geng et al., 2017, Seedat et al., 2022]. Moreover, this provides insight into how irregular observing can achieve a higher expected utility (Section 2.1), as observations could be more informative (hence taken more frequently) when the future state change magnitude is larger—occurring for large $v$.

Furthermore, as in Figure 4 for the Pendulum environment, we observe that AOC observes infrequently the state of the pendulum when it has been swung up to the upright goal state and is balancing it there. This matches our intuition from Section 2.1, as future state changes of $\phi$ the angle of the pendulum are large when the pendulum is not at the equilibrium goal state, thus having $\dot{\phi} > 0$, which is maximally large in the swing up state trajectory phase and necessitates frequent observing, whereas balancing near the top equilibrium point, $\dot{\phi} \approx 0$ necessitates infrequent observing.

On the other hand, quantitatively, as tabulated in Table 4, we make Continuous Planning take the same number of observations as AOC, however, taking them regularly. Since AOC can take better-located observations in time, it outperforms Continuous Planning, achieving a higher reward $\mathcal{R}$. In summary, these experiments demonstrate that actively observing can be superior to planning in continuous time.

**Why is it crucial to actively observe with continuous-time methods, rather than discrete-time methods?** We compare Active Observing Control to Discrete Monitoring, which is the closest discrete counterpart (a discrete ablation of AOC) and look at when the observing times are reached, relative to their reward uncertainty estimates, to determine whether time discretization affects observing times. As shown in Figure 5, we see that Discrete Monitoring takes observations that are delayed, and therefore achieves a lower state reward $\mathcal{R}$. This allows AOC to capture fast-moving states, where missing these can be catastrophic or deadly, as in the case of cancer.

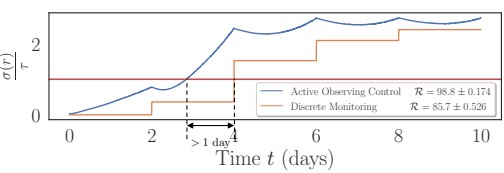

Figure 5: **Reward uncertainty $\sigma(r)$ normalized by the method specific threshold $\tau$ on the cancer environment**—thus the threshold to observe is at 1.0, as indicated by the red horizontal line. We observe that AOC can detect and catch when the uncertainty crosses the threshold, independent of the time discretization. In contrast, Discrete Monitoring suffers from time discretization error and takes a delayed observation, leading to compounding errors and worse performance. Here Discrete Monitoring misses observing a critical test by over a day.

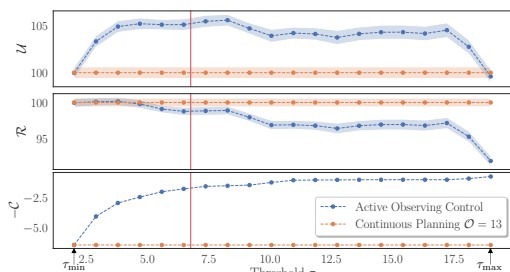

Figure 6: **Normalized utility $\mathcal{U}$ on the cancer environment of Active Observing Control and Continuous Planning**, plotted against changing uncertainty threshold $\tau$. We plot AOC's threshold as used in Table 3 as the red line. AOC maintains a high utility and reward over a wide range of thresholds—whilst using fewer observations as $\tau$ increases, compared to Continuous Planning which takes expensive frequent observations. The threshold $\tau$ is between the minimum feasible $\tau_{\min}$ and maximum feasible $\tau_{\max}$ values.

## 5.3 Sensitivity analysis of $\tau$

Active Observing Control still outperforms Continuous Planning even at variable $\tau$ on the cancer environment, Figure 6. We note that as $\tau$ is increased, this increases the observing times $\Delta_{i+1}$, and hence fewer observations of the state are taken. Although we set $\tau = 6.67$ for the cancer environment, following the procedure in Section 4, we observe robustness to other choices of $\tau$ are also suitable with decreasing reward $\mathcal{R}$; however, AOC still outperforms and with a high utility $\mathcal{U}$. Moreover, we tuned $\tau_{\min}$ so that frequent observations were taken to be at least equal to Continuous Planning, and then $\tau_{\max}$ to take the fewest observations, which is often minimally 1, as we start by taking an observation, or until the entire action trajectory plan is executed, and requires re-planning by taking another observation.

## 6 Conclusion and Future work

This work lays the foundation for the significant yet unexplored real-world problem of continuous-time control whilst deciding when to take costly observations, which has profound implications across diverse fields such as medicine, resource management, and low-power systems. Our key theoretical contribution, verified empirically is that regular observing is not optimal and that irregularly observing can achieve a higher expected utility. To demonstrate the power of active irregular observing, we provided an initial solution using a heuristic threshold based on the variance of reward rollouts in an offline continuous-time model-based model predictive control planner. However, the quest for the optimal method remains an open problem, providing fertile ground for future research.

However, this work is not without limitations. The heuristic threshold, while a robust initial solution in our experiments, may not be optimal or suitable for all scenarios. Moreover, our approach relies on the assumption that the offline dataset provides sufficient coverage of the state-action space to learn a useful enough model of the system dynamics and that the observation observes all the states plus Gaussian noise. Also, in some practical applications, the cost of observations could vary or be unknown.

Navigating these limitations illuminates exciting pathways for future research. Potential directions include: optimizing multiple observations concurrently over a planned action trajectory, jointly optimizing action trajectories and observing times, and theoretically deriving the optimal solution to reveal characteristics of optimal observing policies.

**Acknowledgements**. SH would like to acknowledge and thank AstraZeneca for funding. This work was additionally supported by the Office of Naval Research (ONR) and the NSF (Grant number: 1722516). Moreover, we would like to warmly thank all the anonymous reviewers, alongside research group members of the van der Scaar lab, for their valuable input, comments and suggestions as the paper was developed—where all these inputs ultimately improved the paper.

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

# Appendix

## Table of Contents

**Code.** All code is available at https://github.com/samholt/ActiveObservingInContinuous-timeControl.

## A  Broader Impact Statement

Our work on active observing in continuous-time control presents a novel perspective on decision-making in domains such as medicine, low-power systems, and resource management, with the potential to substantially reduce costs and increase efficiency. However, the adoption of these techniques also presents potential risks and ethical concerns. The dependability of these systems could raise issues if they malfunction or are compromised, potentially leading to harmful decisions in critical fields like healthcare. Hence, robustness, security, and error handling must be prioritized in implementation.

## B  Further Application Examples

The problem of continuously controlling continuous time environments whilst actively deciding when to take costly observations in time is ubiquitous in real-world environments, with some of these applications listed in Table 5.

Table 5: **Applications of continuous control with costly observations.**

| Continuous time environment | Ref. | Continuous control | Costly observation | High stakes |
|---|---|---|---|---|
| Medical cancer chemotherapy treatment | [Geng et al., 2017, Brunereau et al., 2012] | Continuous chemotherapy dosing | Computed tomography scan | Errors can be fatal |
| Biological fish population management | [Yoshioka and Tsujimura, 2020] | Continuous food and temperature control | Fish population survey | Underfeeding or overfishing can lead to extinction |
| Low power communication | [Mastronarde and van der Schaar, 2012, Lin and van der Schaar, 2010] | Continuous channel transmission | Measure the maximum bandwidth | Communication rate errors lead to loss of data |
| Mobile Robotics | [Martinez-Cantin et al., 2007] | Continuous robot control | Measure the robot's position | Errors can lead to damage to the robot or environment |
| Agricultural resource management | [Larson, 1992] | Continuous land allocation for livestock | Measure the health of livestock | Errors can lead to livestock death |
| Nursing surveillance | [Morgan et al., 2005] | Continuous treatment plans | Measure the health of a patient | Errors can be fatal |
| Water pollution monitoring | [Twardowska et al., 2007] | Continuous water treatment | Measure the water quality | Errors can lead to water contamination |

## C  Proof for Proposition 2.1

*Proof.* We will consider the system defined by

$$\frac{\mathrm{d}s}{\mathrm{d}t} = s \tag{2}$$

$$s(0) = \begin{cases} 1 & \text{with probability} \quad 1/2 \\ -1 & \text{with probability} \quad 1/2 \end{cases} \tag{3}$$

$$\sigma_\epsilon = 1 \tag{4}$$

Note that solution to this system can be written as $s(t) = s(0)e^t$, which is independent of any actions that might have been taken. Also, consider the reward function given by

$$r(s, a, t) = \delta(t - T) \cdot \mathbb{1}\{s \cdot a > 0\} \tag{5}$$

Given a history of samples $h = \{(t_j, z(t_j), a(t_j))\}$, a posterior over $s(0)$ can be written as

$$\mathbb{P}(s(0) = 1|h) = \frac{\mathbb{P}(s(0) = 1)\mathbb{P}(h|s(0) = 1)}{\mathbb{P}(h)} = \frac{1}{\mathbb{P}(h)} \prod_{j=1}^{|h|} \frac{1}{\sqrt{2\pi}} e^{-1/2(z(t_j) - e^{t_j})^2} \qquad (6)$$

$$\mathbb{P}(s(0) = -1|h) = \frac{\mathbb{P}(s(0) = -1)\mathbb{P}(h|s(0) = -1)}{\mathbb{P}(h)} = \frac{1}{\mathbb{P}(h)} \prod_{j=1}^{|h|} \frac{1}{\sqrt{2\pi}} e^{-1/2(z(t_j) + e^{t_j})^2} \qquad (7)$$

Using this posterior, we define an auxiliary variable

$$b(h) = \log \mathbb{P}(s(0) = 1|h) - \log \mathbb{P}(s(0) = -1|h) \qquad (8)$$

$$= -\frac{1}{2} \sum_{j=1}^{|h|} (z(t_j) - e^{t_j})^2 + \frac{1}{2} \sum_{j=1}^{|h|} (z(t_j) + e^{t_j})^2 \qquad (9)$$

Then, the optimal decision policy can be written as

$$\pi^*(h, t) = \begin{cases} 1 & \text{if} \quad \mathbb{P}(s(0) = 1|h) \geq 1/2 \iff b(h) \geq 0 \\ -1 & \text{if} \quad \mathbb{P}(s(0) = 1|h) < 1/2 \iff b(h) < 0 \end{cases} \qquad (10)$$

Next, we derive the expected utility of the optimal decision policy given a set of observing times $\{t_j \in [0, T]\}_{j=1}^n$. Denoting with $h_n = \{(t_j, z(t_j), a(t_j))\}_{j=1}^n$ the final history, we have

$$\mathcal{R} = \int_0^T r(s(t), a(t), t) \mathrm{d}t \qquad (11)$$

$$= \int_0^T \delta(t - T) \cdot \mathbb{1}\{s(t) \cdot a(t) > 0\} \mathrm{d}t \qquad (12)$$

$$= \mathbb{1}\{s(T) \cdot a(T) > 0\} \qquad (13)$$

$$= \mathbb{1}\{s(T) \cdot \pi^*(h_n, T) > 0\} \qquad (14)$$

$$= \mathbb{1}\{s(0) = 1 \wedge \pi^*(h_n, T) = 1\} + \mathbb{1}\{s(0) = -1 \wedge \pi^*(h_n, T) = -1\} \qquad (15)$$

$$= \mathbb{1}\{s(0) = 1 \wedge b(h_n) \geq 0\} + \mathbb{1}\{s(0) = -1 \wedge b(h_n) < 0\} \qquad (16)$$

Hence,

$$\mathbb{E}[\mathcal{R}|\{t_j\}_{j=1}^n] = \mathbb{P}(s(0) = 1 \wedge b(h_n) \geq 0|\{t_j\}) + \mathbb{P}(s(0) = -1 \wedge b(h_n) < 0||\{t_j\}) \qquad (17)$$

$$= \mathbb{P}(s(0) = 1|\{t_j\})\mathbb{P}(b(h_n) \geq 0|s(0) = 1, \{t_j\}) \qquad (18)$$

$$+ \mathbb{P}(s(0) = -1|\{t_j\})\mathbb{P}(b(h_n) < 0|s(0) = -1, \{t_j\}) \qquad (19)$$

$$= 1/2 \cdot \mathbb{P}(b(h_n) \geq 0|s(0) = 1, \{t_j\}) + 1/2 \cdot \mathbb{P}(b(h_n) < 0|s(0) = -1, \{t_j\}) \qquad (20)$$

Observing that

$$\mathbb{P}(b(h_n) \geq 0 | s(0) = 1, \{t_j\}) \tag{21}$$

$$= \mathbb{P}\left(-\frac{1}{2}\sum_{j=1}^{b}(z(t_j) - e^{t_j})^2 + \frac{1}{2}\sum_{j=1}^{b}(z(t_j) + e^{t_j})^2 \geq 0 \,\Big|\, s(0) = 1\right) \tag{22}$$

$$= \mathbb{P}\left(-\frac{1}{2}\sum_{j=1}^{b}(\varepsilon(t_j) + s(t_j) - e^{t_j})^2 + \frac{1}{2}\sum_{j=1}^{b}(\varepsilon(t_j) + s(t_j) + e^{t_j})^2 \geq 0 \,\Big|\, s(0) = 1\right) \tag{23}$$

$$= \mathbb{P}\left(-\frac{1}{2}\sum_{j=1}^{b}(\varepsilon(t_j))^2 + \frac{1}{2}\sum_{j=1}^{b}(\varepsilon(t_j) + 2e^{t_j})^2 \geq 0\right) \tag{24}$$

$$= \mathbb{P}\left(\sum_{j=1}^{b}(e^{2t_j} + e^{t_j}\varepsilon(t_j)) \geq 0\right) \tag{25}$$

$$= \mathbb{P}_{\bar{\varepsilon}\sim\mathcal{N}\left(0, \sum_{j=1}^{n} e^{2t_j}\right)}\left(\bar{\varepsilon} > -\sum_{j=1}^{n} e^{2t_j}\right) \tag{26}$$

$$= \mathbb{P}_{\bar{\varepsilon}\sim\mathcal{N}(0,1)}\left(\bar{\varepsilon} > -\sqrt{\sum_{j=1}^{n} e^{2t_j}}\right) \tag{27}$$

$$= F\left(\sqrt{\sum_{j=1}^{n} e^{2t_j}}\right) \tag{28}$$

where $F(x) = \int(1/\sqrt{2\pi})e^{-(1/2)x^2}$ is the cumulative distribution function of the standard normal distribution, and similarly that $\mathbb{P}(b(h_n) \geq 0 | s(0) = -1, \{t_j\}) = F(\sqrt{\sum_{j=1}^{n} e^{2t_j}})$, we obtain

$$\mathbb{E}[\mathcal{U}|\{t_j\}_{j=1}^{n}] = \mathbb{E}[\mathcal{R}|\{t_j\}_{j=1}^{n}] - cn = \frac{1}{2}F\left(\sqrt{\sum_{j=1}^{n} e^{2t_j}}\right) - cn \tag{29}$$

Next, we introduce two classes of observing policies. The first class consists of observing policies that observe regularly in time with an observing interval of $\delta$ such that $\rho^{(\text{reg},\delta)}(h) = \delta$ for all $h$. When enrolled, the observing times for these policies can be written as $t_j = j\delta$ hence

$$\mathbb{E}_{\rho^{(\text{reg},\delta)}}[\mathcal{U}] = \frac{1}{2}F\left(\sqrt{\sum_{j=1}^{\lfloor T/\delta \rfloor} e^{2j\delta}}\right) - c\left\lfloor\frac{T}{\delta}\right\rfloor \tag{30}$$

The second class consists of observing policies that take $\ell$-many observations all at $t = T$ such that

$$\rho^{(\text{irreg},\ell)}(h = \{(t_j, z(t_j), a(t_j))\}_{j=1}^{n}) = \begin{cases} T - t_j & \text{if } n < \ell \\ \infty & \text{if } n \geq \ell \end{cases} \tag{31}$$

When enrolled, the observing times for these policies can be written as $t_j = T$ hence

$$\mathbb{E}_{\rho^{(\text{irreg},\ell)}}[\mathcal{U}] = \frac{1}{2}F(\sqrt{\ell e^{2T}}) - c\ell \tag{32}$$

Crucially, $\rho^{(\text{irreg},\ell)}$ with $\ell \geq 2$ is distinct from all regular observing policies $\rho^{(\text{reg},\delta)}$ with $\delta > 0$, since for no regular observing policy, $t_1 = t_2$. We prove that $\rho^*$ cannot be a regular observing policy by showing that, for each $\rho^{(\text{reg},\delta)}$, there exists at least one $\rho^{(\text{irreg},\ell)}$ with $\ell \geq 2$ that achieves higher expected utility.

For $\delta = 0$, $\mathbb{E}_{\rho^{(\text{reg},\delta)}}[\mathcal{U}] \to -\infty$ whereas $\mathbb{E}_{\rho^{(\text{irreg},\lfloor T/\delta \rfloor)}}[\mathcal{U}]$ is bounded for any $\ell \in \{2, 3, \ldots\}$ (hence $\mathbb{E}_{\rho^{(\text{irreg},\ell)}}[\mathcal{U}] > \mathbb{E}_{\rho^{(\text{reg},\delta)}}[\mathcal{U}]$).

For $\delta \leq T/2$, we have $\lfloor T/\delta \rfloor \geq 2$ and $\mathbb{E}_{\rho^{(\text{irreg}, \lfloor T/\delta \rfloor)}}[\mathcal{U}] > \mathbb{E}_{\rho^{(\text{reg}, \delta)}}[\mathcal{U}]$ since

$$\mathbb{E}_{\rho^{(\text{reg}, \delta)}}[\mathcal{U}] - \mathbb{E}_{\rho^{(\text{irreg}, \lfloor T/\delta \rfloor)}}[\mathcal{U}] \tag{33}$$

$$= \frac{1}{2}F\left(\sqrt{\sum_{j=1}^{\lfloor T/\delta \rfloor} e^{2j\delta}}\right) - c\left\lfloor \frac{T}{\delta} \right\rfloor - \frac{1}{2}F\left(\sqrt{\left\lfloor \frac{T}{\delta} \right\rfloor e^{2T}}\right) + c\left\lfloor \frac{T}{\delta} \right\rfloor \tag{34}$$

$$= \frac{1}{2}F\left(\sqrt{\sum_{j=1}^{\lfloor T/\delta \rfloor} e^{2j\delta}}\right) - \frac{1}{2}F\left(\sqrt{\left\lfloor \frac{T}{\delta} \right\rfloor e^{2T}}\right) \tag{35}$$

$$< \frac{1}{2}F\left(\sqrt{\sum_{j=1}^{\lfloor T/\delta \rfloor} e^{2T}}\right) - \frac{1}{2}F\left(\sqrt{\left\lfloor \frac{T}{\delta} \right\rfloor e^{2T}}\right) = 0 \tag{36}$$

Let the cost per observation be given as

$$c = \frac{1}{2}\min\left\{\frac{1}{2}F\left(\sqrt{2e^{2T}}\right) - \frac{1}{2}F\left(\sqrt{e^{2T}}\right), \frac{1}{4}F\left(\sqrt{e^{2T}}\right) - \frac{1}{4}F(0)\right\} > 0 \tag{37}$$

Then, for $T/2 < \delta \leq T$, we have $\lfloor T/\delta \rfloor = 1$ and $\mathbb{E}_{\rho^{(\text{irreg}, 2)}}[\mathcal{U}] > \mathbb{E}_{\rho^{(\text{reg}, \delta)}}[\mathcal{U}]$ since

$$\mathbb{E}_{\rho^{(\text{reg}, \delta)}}[\mathcal{U}] - \mathbb{E}_{\rho^{(\text{irreg}, 2)}}[\mathcal{U}] \tag{38}$$

$$= \frac{1}{2}F\left(\sqrt{e^{2\delta}}\right) - c - \frac{1}{2}F\left(\sqrt{2e^{2T}}\right) + 2c \tag{39}$$

$$= \frac{1}{2}F\left(\sqrt{e^{2\delta}}\right) - \frac{1}{2}F\left(\sqrt{2e^{2T}}\right) + c \tag{40}$$

$$\leq \frac{1}{2}F\left(\sqrt{e^{2T}}\right) - \frac{1}{2}F\left(\sqrt{2e^{2T}}\right) + c = 0 \tag{41}$$

$$\leq \min\left\{\frac{1}{2}F\left(\sqrt{e^{2T}}\right) - \frac{1}{2}F\left(\sqrt{2e^{2T}}\right), \frac{1}{4}F(0) - \frac{1}{4}F\left(\sqrt{e^{2T}}\right)\right\} + c \tag{42}$$

$$= -2c + c < 0 \tag{43}$$

Finally, for $T < \delta$, we have $\lfloor T/\delta \rfloor = 0$ and $\mathbb{E}_{\rho^{(\text{irreg}, 2)}}[\mathcal{U}] > \mathbb{E}_{\rho^{(\text{reg}, \delta)}}[\mathcal{U}]$ since

$$\mathbb{E}_{\rho^{(\text{reg}, \delta)}}[\mathcal{U}] - \mathbb{E}_{\rho^{(\text{irreg}, 2)}}[\mathcal{U}] \tag{44}$$

$$= \frac{1}{2}F(0) - \frac{1}{2}F\left(\sqrt{2e^{2T}}\right) + 2c \tag{45}$$

$$\leq \min\left\{\frac{1}{2}F(0) - \frac{1}{2}F\left(\sqrt{2e^{2T}}\right), F\left(\sqrt{e^{2T}}\right) - F\left(\sqrt{2e^{2T}}\right)\right\} + 2c \tag{46}$$

$$= -4c + 2c < 0 \tag{47}$$

$\square$

# D   Extended Related Work

In the following, we expand on the existing related work, in Section 3, and we provide additional discussions of the benefits of offline RL, model-based RL, why model predictive control is our preferred action policy, event-based sampling, Semi-MDP methods, and Linear Quadratic Regression (LQR) & Linear Quadratic Gaussian (LQG) methods.

**Why offline RL**. The setting of offline reinforcement learning consists of an agent that learns a dynamics model (i.e., a model of the environment dynamics) from a dataset of previously collected state-action trajectories from a specific environment and the agent is not allowed to interact with the environment [Wu et al., 2019]. For instance, it is challenging to apply *online* RL methods to real-world problem settings, as they rely on expensive trial-and-error approaches performed either online or in a realistic simulator—that is not often possible. In contrast, "offline" model-based RL learns a model of the environment dynamics from a previously collected dataset of state-action trajectories, that *is* often possible. These methods then control the environment to a desired goal state using any available planning method such as training a policy [Fujimoto et al., 2018] or using

a model predictive controller (MPC) [Williams et al., 2016]. Specifically, both model-free [Kumar et al., 2019, 2020, Fujimoto and Gu, 2021] and model-based [Kidambi et al., 2020, Wang et al., 2021] approaches have been proposed for offline RL, where in general model-based methods have shown to be more sample efficient than model-free methods [Moerland et al., 2020].

**Why model-based reinforcement learning?** Core to Model-based RL methods is the desire to create policies for real world tasks in the offline setting where the true environment dynamics are unknown and rather learn an approximate dynamics model from an offline dataset of state-action trajectories from the environment. A key benefit of model-based reinforcement learning is that it can be significantly more sample efficient compared to model-free RL methods [Lutter et al., 2021]— where model-free methods can require millions or billions of interactions with the environment to learn a good policy. Another key benefit is that learning a dynamics model of the environment enables planning with that dynamics model to optimize actions, such as in using a model predictive control planner, rather than learning a specific policy.

Principally, the dynamics model is independent of the reward and therefore allows changing reward of the planning policy—enabling a planning-based policy to easily adapt to changing goal states/tasks on the fly at run-time. Conversely, a policy that was solely trained for a particular task would have to be re-trained, or a conditional policy trained when adapting goals/tasks [Lutter et al., 2021]. This is realistic as real-world continuous-time control settings of the dynamics model (e.g., often the physics/biological process of an environment [Holt et al., 2022a]) is independent of the reward function. Importantly model-based RL for conventional control tasks has been shown to be more sample-efficient than model-free methods [Wang et al., 2019, Moerland et al., 2020]. Historically, model-free methods have demonstrated expert performance on challenging tasks [Mnih et al., 2015, Lillicrap et al., 2015], and was later shown that proper tuning of model-based methods have much higher sample efficiency Ha and Schmidhuber [2018], Janner et al. [2019] and can achieve expert performance with probabilistic dynamics models on continuous control tasks [Chua et al., 2018].

**Model predictive control (MPC).** Core to model predictive control (MPC) is having a good dynamics model, historically using simple, first principle derived (often linear), known dynamics models [Richalet et al., 1978, Salzmann et al., 2022]. Recently, MPC Model Predictive Path Integral (MPPI) [Williams et al., 2017] is a zeroth-order particle-based (action) trajectory optimizer [Williams et al., 2017] method that can use a general, non-differentiable nonlinear dynamics model with a defined complex reward function (or cost function).

Particularly, Williams et al. [2017] demonstrated that it was suitable to be used with a learned neural network dynamics model—which was used for the task of driving a vehicle around a dirt track in aggressive, actuator saturating maneuvers. In general, MPC is often computationally infeasible for optimizing actions up to long time horizons, therefore are often used to optimize actions up to a fixed time horizon, which is re-computed iteratively when needed. A key benefit of planning, and MPC, is that it can incorporate state constraint feasibility into its plans as well as easily changing the reward function for a changing goal/task at run-time. We specifically use this MPC method in this paper, as it is state-of-the-art [Wang et al., 2022] for performing MPC with a learned dynamics model, improving upon the widely used cross-entropy method (CEM) MPC planner [Kobilarov, 2012].

**Hybrid MPC:** Combining a powerful MPC planner with a learned policy is another exciting area developing [Argenson and Dulac-Arnold, 2020]. These existing hybrid works, work on only discrete-time environment dynamics models. These include MBOP [Argenson and Dulac-Arnold, 2020], TD-MPC [Hansen et al., 2022] and DADS [Sharma et al., 2019]. We highlight that these hybrid methods are unable to determine when to observe in continuous-time and only determine discrete actions to execute.

**Sensing** approaches have been proposed of when to optimally take an observation in both discrete time [Jarrett and Van Der Schaar, 2020] and continuous time [Alaa and Van Der Schaar, 2016, Barfoot et al., 2014]—where their goal is to identify an underlying state. However, these approaches cannot also continuously control the system. In contrast, Active Observing Control seeks to both actively observe and control the system. Besides deciding when to take an observation, some work on sensing focuses on when to stop taking observations [e.g. Hüyük et al., 2023] or what kind of observations to take [e.g. Curth et al., 2023]. Unlike these works, we focus exclusively on the timing of observations.

**Planning** approaches only focus on optimal decision policies $\pi$, and therefore observing has to be provided by a schedule, which is often assumed to be at a regular constant interval in time

$\Delta_{i+1} = \Delta_i$—that is, observations are not actively planned [Yildiz et al., 2021]. Within planning approaches, there exist many discrete-time approaches [Chua et al., 2018, Mnih et al., 2013, Williams et al., 2017, Sun et al., 2023] and recently more continuous-time approaches [Du et al., 2020, Yildiz et al., 2021, Holt et al., 2023]—where these use a continuous-time dynamics model, e.g. [Chen et al., 2018, Holt et al., 2022b]. Specifically, Yildiz et al. [2021] presented a seminal online continuous-time model-based RL algorithm, leveraging a continuous-time dynamics model that can predict the next state at an arbitrary time $s(t)$. However, all these methods are *unable* to plan when to take the next observation, in contrast to Active Observing Control which *is* able to determine when to take the next observation whilst planning an action trajectory.

**Monitoring** approaches consist of both a decision policy $\pi$ and an observing policy $\rho$; however, existing approaches only consider the discrete-time and discrete-state setting with observation costs [Sharma et al., 2017, Nam et al., 2021, Aguiar et al., 2020, Bellinger et al., 2020a, Krueger et al., 2020, Bellinger et al., 2020b]. In particular, Krueger et al. [2020] proposed that even in a simple setting of a discrete-state multi-armed bandit, computing the optimal time to take the next observation is intractable—therefore, they must rely on heuristics for their observing policy[1].

Broadly, discrete-time monitoring methods use a discrete-time model and propagate the last observed state, often appending either a flag or a counter to the state-action tuple to indicate the time since an observation was taken [Aguiar et al., 2020, Bellinger et al., 2020b]. However, these methods cannot be applied to continuous-time environments or propagate the predicted current state and its associated prediction interval of the uncertainty associated with the state estimate. Moreover, training a policy [Aguiar et al., 2020] that decides at each state whether to observe it or not, requires a discretization of time. Whereas Active Observing Control, which determines the continuous-time interval of when to observe the next state, does not require any discretization of time, and hence it is a continuous-time method and avoids compounding time discretization errors.

One approach exists that we term *semi-continuous monitoring*, where Huang et al. [2019] proposes a discrete-state, constant action policy, that determines when to take an observation in continuous time of a controlled Markov Jump Process. However, this approach is limiting, as it assumes actions are constant until the next sample of the state is taken [Ni and Jang, 2022]—which is clearly suboptimal, uses discrete states, and assumes a Markov Jump Process [Gillespie, 1991]. Instead, Active Observing Control is fully continuous in both states and observing times—controlled by an action trajectory $a$, giving rise to *smoothly-varying* states.

**Event-based sampling** a similar related work area in the control community [Åström and Bernhardsson, 1999, Åström, 2008] , which addresses the similar problem of controlling a system, of only taking a full state observation when an external input *event* occurs, and then providing a control action. To create this *event*, it *assumes part of the state is continuously observed*, and often an *event* is defined when this part of the state (or a function of it) crosses a fixed threshold (e.g., a physical sensor crossing a threshold) [Vasyutynskyy and Kabitzsch, 2010]. This finds multiple uses in problems such as electronic analog control systems for audio and mobile telephone systems [Åström and Bernhardsson, 1999], and battery capacity optimization in wireless devices [Vasyutynskyy and Kabitzsch, 2010]. However, this is different from our proposed problem of continuous-time control whilst deciding when to take costly observations as (1) Event-based sampling *assumes part of the state is continuously observed*. Often in our environments, it is not feasible to observe part of the state continuously (e.g., imaging a cancer tumor volume or performing a medical test) or it is prohibitively expensive to continually observe at a high-frequency part of the state (similar to Continuous Planning approaches). (2) The *event* input (the time to take the next observation) is given as a control input to the agent. However, it is precisely the more difficult problem we tackle of coming up with an observing policy that decides *when* to take the next observation. (3) The *event* is often defined by a human in advance and is a function of the current partial state. General environments may not have partial state spaces that are predictive of when to observe next, such as a robotic navigation task in two dimensions and only continuously observing one dimension.

---

[1]Krueger et al. [2020], focus on the simpler problem of multi-armed bandits (MAB), where there is a cost to observe the reward. In the MAB setting, it is possible to formulate it as an RL PO-MDP with one state. The underlying static state (the fixed reward distributions of each bandit) is unknown and must be determined by taking successive costly observations (paying a cost to observe a reward). Therefore, Krueger et al. [2020]'s statement that the optimal algorithm for their simple MAB setting is intractable applies to the problem of active observing in continuous-time control.

**Semi-MDP methods** is another similar related field, where it extends the MDP problem to include *options* that define temporally abstracted action sequences [Sutton, 1998]. However, some distinct differences prevent using a semi-MDP to address our continuous-time control whilst deciding when to take costly observations problem. These differences include (1) Semi-MDP is still an underlying discrete MDP with discrete state transitions, whereas our problem formulation focuses on continuous-time systems that can handle continuous actions and states. (2) Semi-MDP formulation does not involve an observation cost, whereas our problem formulation does.

**LQR/LQG methods**. Similar related methods from control exist, as Zhang et al. [2022] propose a discrete planning method that works only for *linear* quadratic regulator (LQR) systems and optimizes the observing frequency for a given total budget of observations. Moreover, Huang and Zhu [2020] addresses an infinite horizon discrete-time linear quadratic Gaussian (LQG) control problem with observation costs. This is notably different from the formulation proposed for the problem of continuous-time control with observation costs as (1) Huang and Zhu [2020] only applies to *linear systems* that are *discrete in time* and have an *infinite time horizon*; our formulation applies to *non-linear systems* that are *continuous in time* and have a *fixed time horizon*. (2) Huang and Zhu [2020] assume their full system dynamics model is *known*; ours makes no such assumptions and only assumes access to an offline collected dataset of (possibly irregular in time) state-action trajectories to learn a dynamics model, which is more applicable to real-world environments.

## E   MPC MPPI Pseudocode and Planner Implementation Details

To plan an optimal action trajectory $a$, we specify that any model predictive controller (MPC) planner can be used—that can optimize the action trajectory up to a specified time horizon $H \in \mathbb{R}_+$. We opt to use an MPC planner, as it optimizes an entire action trajectory $a$ up to $H$, does not require the probabilistic dynamics model $\hat{f}_\theta$ to be differentiable (it uses no gradients), and the reward function $r$ can be changed on the fly—allowing changing goals/tasks at run-time. Moreover, model-based RL with MPC planners has achieved comparable performance to model-free RL methods [Chua et al., 2018, Lutter et al., 2021]. We use our probabilistic dynamics model $\hat{f}_\theta$, with the MPC planner of Model Predictive Path Integral Control (MPPI), a zeroth-order particle-based trajectory optimizer [Williams et al., 2017], due to its competitive performance [Wang et al., 2022].

To optimize the action trajectory $a$ up to a time horizon $H$, it discretizes $H$ into smaller constant action time intervals $\delta_a \in \mathbb{R}_+$ which are then optimized—where there are $K = \lceil \frac{H}{\delta_a} \rceil \in \mathbb{Z}_+$ time steps in $H$— i.e., $t^{(k)} = t_i + k\delta_a, k \in \{0, \ldots, K-1\}$. It forward simulates a number of parallel rollouts $G \in \mathbb{Z}_+$, where the next state estimate at each time step is simulated as $z(t^{(k+1)}) = \mu_*(z(t^{(k)}), a(t^{(k)}), \delta_a)$ recursively. This requires a state estimate $z(t_i)$ to plan from; therefore, we recompute the planned trajectory when we observe a new sample of the state. This optimizes the action trajectory by

$$a^*(t^{(\cdot)}) = \underset{a(t^{(\cdot)})}{\arg\max} \, \mathbb{E}_z \left[ \sum_{k=0}^{K-1} r(z(t^{(k)}), a(t^{(k)}), t^{(k)}) \right]$$

MPPI is also a Monte Carlo based sampler, and thus increasing the number of rollouts improves the input trajectory optimization, however, scales the run-time complexity as $O(GK)$. The standard MPPI algorithm [Williams et al., 2017] is used with our probabilistic dynamics model $\hat{f}_\theta$. We articulate the MPPI pseudocode for the action trajectory optimizer in Algorithm 2.

**Algorithm 2** MPPI-Trajectory-Optimization

---

**Input:** State observation $z(t_i)$, Pre-trained probabilistic dynamics model $\hat{f}_\theta$, Reward function $r$, Time Horizon $H$, Action time interval $\delta_a$, Number of parallel rollouts $G$, Noise covariance $\Sigma$, Hyper parameters $\lambda$, Action max $a_{\max}$, Action min $a_{\min}$.

$K \leftarrow \frac{H}{\delta_a}$

$\boldsymbol{R}^{(G)} \leftarrow \mathbf{0}^{(\mathbf{G})}$            $\triangleright$ This holds our $G$ trajectory returns.

$\boldsymbol{A}^{(G,K)} \leftarrow \mathbf{0}^{(\mathbf{G},\mathbf{K})}$            $\triangleright$ This holds our $G$ action trajectories of length $K$.

$\boldsymbol{A}'^{(G,[0:K-1])} \leftarrow \mathbf{0}^{(\mathbf{G},\mathbf{K})}$        $\triangleright$ This holds $G$ action trajectories of length $K$ that are perturbed by noise.

$\boldsymbol{\varepsilon}^{(G,K)} \leftarrow \mathbf{0}^{(\mathbf{G},\mathbf{K})}$            $\triangleright$ This holds the generated scaled action noise.

**for** $g = 0, \ldots, G-1$ **:**
    $\boldsymbol{z}(t^{(0)}) \leftarrow z(t_i)$            $\triangleright$ Sample $G$ trajectories over the horizon $K$.
    **for** $k = 0, \ldots, K-1$ **:**
        $\varepsilon^{(g,k)} \leftarrow \mathcal{N}(\mathbf{0}, \boldsymbol{\Sigma})$            $\triangleright$ Sample action noise.
        $\boldsymbol{A}'^{(g,k)} \leftarrow \boldsymbol{A}^{(g,k)} + \boldsymbol{\epsilon}^{(g,k)}$            $\triangleright$ Perturb action by noise.
        $\boldsymbol{A}'^{(g,k)} \leftarrow \min(\max(\boldsymbol{A}'^{(g,k)}, -1), +1)$ $\triangleright$ Clip normalized perturbed noise (to bound actions to their limits).
        $\varepsilon^{(g,k)} \leftarrow \boldsymbol{A}'^{(g,k)} - \boldsymbol{A}^{(g,k)}$      $\triangleright$ Update noise after bounding, so we do not penalize clipped noise.
    **for** $k = 0, \ldots, K-1$ **:**
        $\boldsymbol{z}(t^{(k+1)}) \leftarrow \mu_*(\boldsymbol{z}(t^{(k)}), \boldsymbol{A}'^{(k)} \cdot \boldsymbol{a}_{\max}, \delta_a)$ $\triangleright$ Sample next state from pre-trained probabilistic dynamics model $\hat{f}_\theta$.
        $\boldsymbol{R}^{(g)} \leftarrow \boldsymbol{R}^{(g)} + r(\boldsymbol{z}(t^{(k+1)}), \boldsymbol{A}^{(g,k)}) - \lambda \boldsymbol{A}^{(g,k)T} \Sigma^{-1} \boldsymbol{\varepsilon}^{(g,k)}$ $\triangleright$ Accumulate the current state reward.
    $\kappa \leftarrow \min_g [\boldsymbol{R}^{(g)}]$
    $\boldsymbol{T}^{(k)} = \dfrac{\sum_{g=0}^{G-1} \exp\left(\frac{1}{\lambda}(\boldsymbol{R}^{(g)}-\kappa)\right) \boldsymbol{\varepsilon}^{(g,k)}}{\sum_{g=0}^{G-1} (\frac{1}{\lambda}(\boldsymbol{R}^{(g)}-\kappa))} \cdot \boldsymbol{a}_{\max}, \forall k \in [0, K-1]$      $\triangleright$ Generate the return-weighted trajectory update.
**Return:** $\boldsymbol{T}$

---

# F    Active Observing Control Planning Algorithm Pseudocode

The Active Observing Control planning algorithm pseudocode is outlined in Algorithm 3. In particular, $\delta_t$ is the continuous search (root finding algorithm) tolerance that is used when searching with binary search for the continuous time duration that the computed action trajectory can be followed for, which occurs when the standard deviation of the reward crosses the threshold $\tau$. We note that all numerical root-finding algorithms generally involve a search tolerance or a similar stopping criterion. The $\delta_t$ tolerance ensures that the binary search algorithm stops evaluations once a solution time is found that is "close enough" to the actual true value. Another way to view the search tolerance $\delta_t$, is that the numerical precision that the search value is correct up to.

---

**Algorithm 3** Active Observing Control Policy

---

**Input:** State observation $z(t_i)$, Pre-trained probabilistic dynamics model $\hat{f}_\theta$, Reward uncertainty threshold $\tau$, Reward function $r$, Time Horizon $H$, Search tolerance $\delta_t$, Action time interval $\delta_a$.

$a \leftarrow \text{MPC}(\hat{f}_\theta, z(t_i), r, H)$  ▷ Plan action trajectory.
**for** $k = 0, \ldots, K - 1$ :
   $t^{(k)} \leftarrow t_i + k\delta_a$
   **for** $p \leftarrow 1, \ldots, P$ :
     $z_p(t^{(k+1)}) \sim \mathcal{N}(\mu_*(z(t^{(k)}), a(t^{(k)}), \delta_a), \sigma_*^2(z(t^{(k)}), a(t^{(k)}), \delta_a))$
   **if** $\sqrt{\mathbb{V}_p[r(z_p(t^{(k+1)}), a(t^{(k+1)}))]} > \tau$ :
     $t^{(\text{lower})} \leftarrow t^{(k)}, t^{(\text{upper})} \leftarrow t^{(k+1)}$
     **while** $t^{(\text{upper})} - t^{(\text{lower})} > \delta_t$ :
       $t^{(\text{mid})} \leftarrow (t^{(\text{upper})} + t^{(\text{lower})})/2$
       **for** $p = 1, \ldots, P$ :
         $z_p(t^{(\text{mid})}) \sim \mathcal{N}(\mu_*(z(t^{(k)}), a(t^{(k)}), t^{(\text{mid})} - t^{(k)}), \sigma_*^2(z(t^{(k)}), a(t^{(k)}), t^{(\text{mid})} - t^{(k)}))$
       **if** $\sqrt{\mathbb{V}_p[r(z_p(t^{(\text{mid})}), a(t^{(\text{mid})}))]} > \tau$ :
         $t^{(\text{upper})} \leftarrow t^{(\text{mid})}$
       **else** :
         $t^{(\text{lower})} \leftarrow t^{(\text{mid})}$
     **Return** $a(t) \forall t \in [t_i, t^{(\text{mid})})$
**Return** $a(t) \forall t \in [t_i, t_K)$

---

## G   Active Observing Control Run-time Complexity

Active Observing Control's run-time complexity of Algorithm 3 is $O(GK + P(K + W))$ where $W = \lceil(\log(\delta_a) - \log(\delta_t))/\log(2)\rceil \in \mathbb{Z}_+$. As $W$ is determined by a continuous binary real line search, on an interval of $\delta_a$ up to a resolution of $\delta_t$.

*Proof.* By definition binary search halves the search interval $\delta_a$ every iteration therefore

$$(1/2)^W \delta_a < \delta_t \tag{48}$$

$$(1/2)^W < \frac{\delta_t}{\delta_a} \tag{49}$$

$$-W\log(2) < \log(\delta_t) - \log(\delta_a) \tag{50}$$

$$W > \frac{\log(\delta_t) - \log(\delta_a)}{\log(2)} \tag{51}$$

$$W = \lceil(\log(\delta_a) - \log(\delta_t))/\log(2)\rceil \tag{52}$$

$\square$

Empirically we chose $P = 10G$, $W = 4$, $G = 10,000$, $K = 20$ for all experiments—therefore the dominating scaling parameter is $G$ the number of particle rollouts. We observe in Table 6 that AOC takes approximately $2.4\times$ longer to plan, which includes both planning the actions and when to observe, compared to Continuous Planning which only plans actions (an MPC step). In the case of the Cancer environment, it can take fewer observations and therefore spend less time planning overall. Moreover, AOC is still practical in environments that require fast decision-making, i.e., robotics, as AOC can still produce a plan in less time than the planning action interval $\delta_a$ for the Cartpole environment, as shown in Table 7. Therefore, AOC is a practical method that can be utilized across different environments (planning cancer treatment plans with high accuracy, whilst being fast enough to control continuous robots). Furthermore, we note that MPC has demonstrated scaling to high dimensional environments (Chua et al. [2018] with $G = 20$, Lutter et al. [2021] with G=500).

Table 6: Normalized utilities $\mathcal{U}$, rewards $\mathcal{R}$, observations $\mathcal{O}$, average time taken to plan policy method $\mathcal{T}^{(\text{Average})}$, total time spent planning policy in episode $\mathcal{T}^{(\text{Total})}$ for the Cancer environment—using the same normalization as in Table 3. Even though Active Observing Control takes $2.4\times$ longer to plan both the actions and the observing time $\mathcal{T}^{(\text{Average})}$, compared to Continuous Planning that only plans the actions—AOC can achieve less total planning time $\mathcal{T}^{(\text{Total})}$ (hence less total compute) because it takes fewer observations in total—here Continuous Planning takes $1.5\times$ longer total planning time $\mathcal{T}^{(\text{Total})}$.

| | Cancer | | | | |
|---|---|---|---|---|---|
| Policy | $\mathcal{U}$ | $\mathcal{R}$ | $\mathcal{O}$ | $\mathcal{T}^{(\text{Average})}$ (ms) | $\mathcal{T}^{(\text{Total})}$ (ms) |
| Active Observing Control | 105 | 98.8 | 3.4 | 31.1 | 105 |
| Continuous Planning | 100 | 100 | 13 | 12.8 | 166 |

Table 7: Normalized utilities $\mathcal{U}$, rewards $\mathcal{R}$, observations $\mathcal{O}$, average time taken to plan policy method $\mathcal{T}^{(\text{Average})}$, total time spent planning policy in episode $\mathcal{T}^{(\text{Total})}$ for the Cartpole environment—using the same normalization as in Table 3. Even though Active Observing Control takes $1.7\times$ longer to plan both the actions and the observing time $\mathcal{T}^{(\text{Average})}$, compared to Continuous Planning that only plans the actions—both policies can be run at real-time as the action interval $\delta_a = 100$ ms for the Cartpole environment—making AOC a practical method.

| | Cartpole | | | | |
|---|---|---|---|---|---|
| Policy | $\mathcal{U}$ | $\mathcal{R}$ | $\mathcal{O}$ | $\mathcal{T}^{(\text{Average})}$ (ms) | $\mathcal{T}^{(\text{Total})}$ (ms) |
| Active Observing Control | 140 | 100 | 43.2 | 24.9 | 1073 |
| Continuous Planning | 100 | 100 | 50 | 14.0 | 700 |

# H  Practical Guidelines to Select $c$

The observation cost $c$ should be decided based on the real-world application at hand and include any human preferences in that application if applicable. In real-world systems (with resource constraints), this cost might correspond to actual monetary cost, computational expense, or energy consumption. Further, if this cost involves a human, for example, a patient receiving medical treatment, they may have preferences for treatment, their health impact (e.g., chemotherapy side effects), and or the number, frequency, and timing of treatments that could be included in the cost $c$. Also, there can often be a trade-off between control performance and the number of observations taken, and one way to trade this off is to tune $c$.

# I  Environment Selection and Details

In the following we discuss our reasoning for why we selected the continuous-time environments. Principally, a continuous-time environment is defined by an underlying differential equation (DE) system (e.g., physics or biological system), which allows for the environment state trajectories to be sampled and simulated at any continuous-time, which is unlike discrete environments [Yildiz et al., 2021, Brockman et al., 2016]. We used an ordinary differential equation solver [Virtanen et al., 2020] to simulate all environments, using an Euler solver at a time resolution of $\delta_{\text{sim}}$ as indicated by the environments parameters. We selected the standard continuous-time control environments from the ODE-RL suite [2] [Yildiz et al., 2021], which consists of three well known environments: Pendulum, CartPole, and Acrobot. Furthermore, as detailed in the following, we implemented a Cancer environment that uses a simulation of a Pharmacokinetic-Pharmacodynamic (PK-PD) model of lung cancer tumor growth [Geng et al., 2017] under continuous dosing treatment of chemotherapy and radiotherapy. Where the same underlying model has also been used by others [Lim, 2018, Bica et al., 2020, Seedat et al., 2022]. We further adapt all the environments to have a fixed cost when taking a sample of the state. To learn a continuous-time dynamics model, it is a standard assumption to assume the dataset of state-action trajectories is sampled irregularly in time [Yildiz et al., 2021,

---

[2]Where the ODE-RL suite of the environments used can be downloaded freely from `https://github.com/cagatayyildiz/oderl`.

Chen et al., 2018]. First let us justify the reasoning behind our choice of observing irregularly in time state-action trajectories from these environments and why existing offline datasets are not suitable.

**Why we cannot use an offline dataset of agent trajectories from a discrete-time environment.** There exist standard discrete-time offline datasets [Fu et al., 2020] of state-action trajectories of agents interacting with discrete-time environments—where the state-action trajectories are sampled at regular time intervals $\Delta_{j+1} = \Delta_j$. Let us hypothetically imagine that there exists a regularly-sampled $\Delta_{j+1} = \Delta_j$ in time state-action trajectory offline dataset. Is it possible to sample the trajectories irregularly? Two approaches come to mind. 1) One could imagine using some form of interpolation (e.g., splines or similar) to interpolate to irregular time steps between states and actions, however doing so would lead to errors in the sampled state-action trajectories in comparison to the true irregularly sampled state-action trajectories at those non-uniform time points. These errors could compound over a dynamics model being trained on these; therefore, we highlight that this approach is unsuitable. 2) Another approach would be to start with a regularly sampled state-action trajectory and then sub-sample state-action times from that regular grid of collected times. However, again this approach is unsuitable, as often environments are captured at run-time with a particular regular observation time interval $\Delta$—where only observing discrete multiples of this (i.e., $n\Delta$, $n \in \mathbb{Z}_+$) would lead to gaps between observations, where the mean of the observation intervals would be larger than that of the environments nominal run-time observation interval $\bar{\Delta}_{\text{Sub-sample}} > \bar{\Delta}_{\text{Original Trajectory}}$. Therefore, we note that this becomes a different problem, as there is less information in the state-action trajectories with large observation interval gaps.

**Offline dataset generation** $\mathcal{D}$**.** Rather, to mitigate both of these issues above (1,2) we prefer to collect an offline dataset ourselves by observing irregularly in time state-action trajectories, where the time interval between the state-action time points is sampled from an exponential distribution $\Delta_{i+1} \sim \text{Exp}(\bar{\Delta})$, with a mean of $\bar{\Delta} = \delta_a$ seconds, and state-action values are randomly sampled, collecting a total of $1e6$ samples [Yildiz et al., 2021]. Furthermore, to increase realism, we add Gaussian noise to observations taken, that of $\mathcal{N}(0, \sigma_\epsilon^2)$ with standard deviation $\sigma = 0.01$.

In all environments the actions are continuous and bounded to a defined range $[a_{\min}, a_{\max}]$. Here we assume a given state $s(t)$ is composed of the position state $q$ and their respective velocities $\dot{q}$, i.e., $s(t) = \{q(t), \dot{q}(t)\}$. Furthermore, each environment uses the reward function of the exponential of the negative distance from the current state to the goal state $q^*$, whilst also penalizing the magnitude of action, and we assume that we are given this reward function when planning—as we often know the desired goal state $q^*$ and our current state $q$. Therefore, the reward function for the environments has the following form:

$$r(\{q(t), \dot{q}(t)\}, a(t)) = \exp\left(-||q(t) - q^*||_2^2 - b||\dot{q}(t)||_2^2\right) - v||a(t)||_2^2 \tag{53}$$

Where $b$ and $v$ are specific environment constants [Yildiz et al., 2021]. Specifically, when we use our MPC planner we observe that it plans better without the exponential operator, and therefore remove it, and use the following reward function throughout, $r(\{q(t), \dot{q}(t)\}, a(t)) = -||q(t) - q^*||_2^2 - b||\dot{q}(t)||_2^2 - v||a(t)||_2^2$. Yildiz et al. [2021] set the environments parameters of $b, v$ to penalize large values and enforce exploration from trivial states, and we use their same values which are also tabulated in Table 8. For all environments we assume a fixed cost of $c = 50$ when observing a sample of the state.

Table 8: Environment specification parameters.

| Base Environment | $b$ | $v$ | $a_{\max}$ | $s_{\text{Init}}$ | $q^*$ |
|---|---|---|---|---|---|
| Pendulum | $1e-2$ | $1e-2$ | $[2]$ | $[0.1, 0.1]$ | $[0, L]$ |
| Cartpole | $1e-2$ | $1e-2$ | $[3]$ | $[0.05, 0.05, 0.05, 0.05]$ | $[0, 0, L]$ |
| Acrobot | $1e-4$ | $1e-2$ | $[4, 4]$ | $[0.1, 0.1, 0.1, 0.1]$ | $[0, 2L]$ |
| Cancer | $1e-3$ | $1e-2$ | $[5, 2]$ | $[1138, 0]$ | $[0, 0]$ |

### I.1   ODE-RL Suite

The starting state, for all these tasks, is hanging down, and the goal is to swing up and stabilize the pole(s) upright [Yildiz et al., 2021] in each environment.

Specifically, we set $\delta_a = 0.1$ seconds, and simulate each ODE system at a time resolution of $\delta_{\text{sim}} = 0.01$ seconds, up until the episode length of $T = 5$ seconds. The goal states for these

ODE-RL suite environments are when the pole(s), each of length $L$ are fully upright, such that their $x, y$ co-ordinates of the tip of the pole reach the goal state. That is where $q^*$ is: $[0, L]$ for the Pendulum environment, $[0, 0, L]$ for the Cartpole environment—where the additional 0 is zero for the cart's $x$ location and in Acrobot is $[0, 2L]$ as there are two poles connected to each other. Furthermore, upon restarting the environment the initial state $s$ is sampled from the uniform distribution of $s_0 \sim \mathcal{U}[-s_{\text{Init}}, s_{\text{Init}}]$ [Yildiz et al., 2021], then the $\theta$ states are added with set angle such that the pole(s) are pointing downwards (i.e., Cartpole $\theta' = \theta_{\text{Init}} + \pi$).

In the following we describe each of the ODE-RL environments introducing the environment with a screenshot figure.

### I.1.1 Cartpole (swing up) Environment

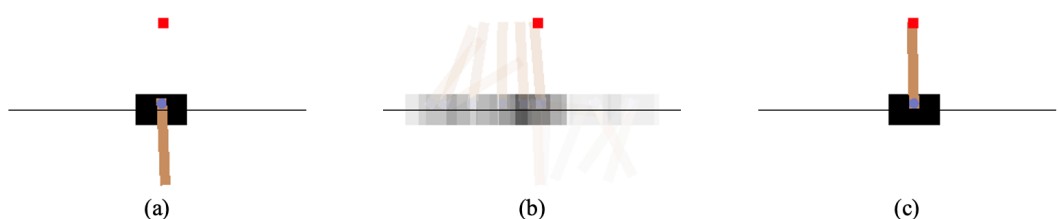

|          (a)          |          (b)          |          (c)          |

Figure 7: Screen shots of the Cartpole environment. The task is to swing up a pole attached to a cart that can move horizontally along a rail. In the following we see: (a) the starting downward state with an additional small amount of perturbation, (b) the optimal trajectory solution found by a policy that scores $\mathcal{R} \approx 100\%$ including our Active Observing Control policy and (c) the final goal state that has been reached, that is, to swing up the pole and stabilize it upwards—which is a challenging control task. We note that the control actuator is bounded and limited, and the force is such that the Cartpole cannot be directly swung up—rather it must gain momentum through a swing and then stabilize this swing to not overshoot when stabilizing the pole upwards in the goal position, as indicated when the tip of the pole reaches the centre of the red target square. Furthermore, we note this environment is an underactuated system.

We can see in Figure 7, an illustration of the starting state Figure 7 (a) with a small perturbed random initial start. Here a pole is attached to an un-actuated joint to a cart that moves along a frictionless track [Barto et al., 1983]. The pendulum starts in the downward position Figure 7 (a) and the goal is to swing the pendulum upwards and then balance the pole upright by applying forces to the left or right horizontal direction of the cart. This environment has the state of $[x, \dot{x}, \theta, \dot{\theta}]$ and a corresponding observation of $[x, \dot{x}, \cos(\theta), \sin(\theta), \dot{\theta}]$, where $\theta \in (-\pi, \pi)$ is measured from the upward vertical of the pole. We note that this environment is an underactuated system, as it has two degrees of freedom $[x, \theta]$, however only the carts position is actuated, leaving $\theta$ indirectly controlled. Additionally, for this Cartpole environment only, created a more competitive Random policy than randomly executing actions, that of applying no actions to keep the pole in the same starting position. This achieves a significantly higher reward than a pure Random policy, as the Cartpole environment is $x \in \mathbb{R}$ unbounded—and a Random policy leads to random drift that significantly moves away from the goal position during the episode length, i.e., $|x| \gg 0$.

### I.1.2 Pendulum Environment

We can see in Figure 8, an illustration of the starting state Figure 8 (a) with a small perturbed random initial start. Here a pole (pendulum) is attached to a fixed point at one end with the other end being free [Barto et al., 1983, Yildiz et al., 2021]. The pendulum starts in the downward position Figure 8 (a) and the goal is to swing the pendulum upwards and then balance the pole upright by applying torques about the fixed point, as indicated in the Figure 8 with a visualization showing the torque direction and magnitude based on the size of the arrow. This environment has the state of $[\theta, \dot{\theta}]$ and a corresponding observation of $[\sin(\theta), \cos(\theta), \dot{\theta}]$.

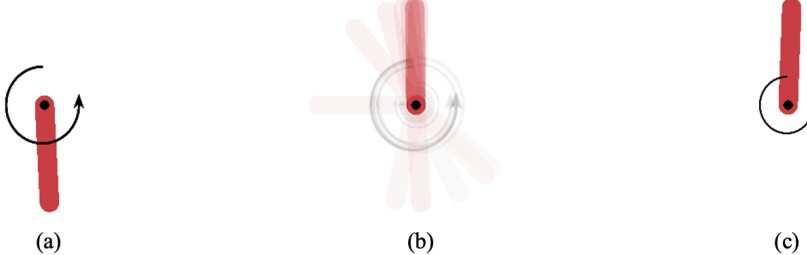

|  (a) | (b) | (c) |

Figure 8: Screen shots of the Pendulum environment. The task is to swing up the pole (pendulum). In the following we see: (a) starting downward state with an additional small amount of perturbation, (b) the optimal trajectory solution found by a policy that scores $\mathcal{R} \approx 100\%$ including our Active Observing Control policy and (c) the final goal state that has been reached, that is, to swing up the pole and stabilize it upwards. We note that the control actuator is bounded and limited, and the force is such that the Pendulum cannot be directly swung up—rather it must gain momentum through a swing and then stabilize this swing to not overshoot when stabilizing the pole upwards in the goal position.

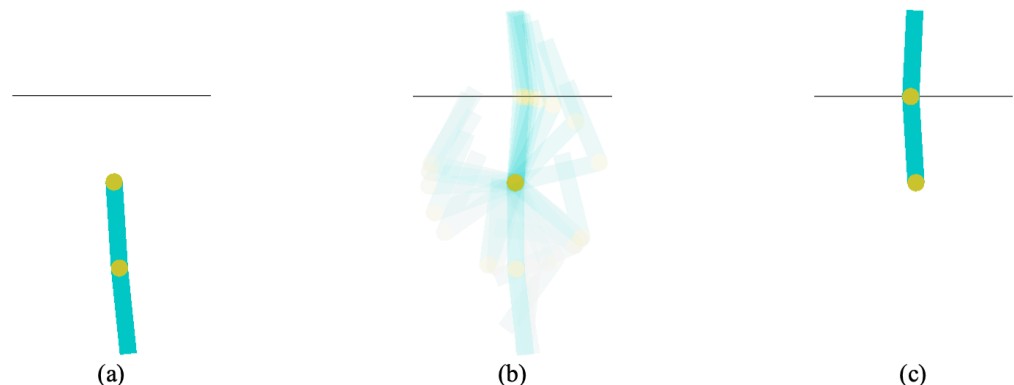

|  (a) | (b) | (c) |

Figure 9: Screen shots of the Acrobot environment. The task is to swing up the 2-link pendulum. In the following we see: (a) starting downward state with an additional small amount of perturbation, (b) the optimal trajectory solution found by a policy that scores $\mathcal{R} \approx 100\%$ including our Active Observing Control policy and (c) the final goal state that has been reached, that is, to swing up the 2-link pendulum and stabilize it upwards. We note that the control actuator is bounded and limited, and the force is such that the 2-link pendulum cannot be directly swung up—rather it must gain momentum through a 2-link swing and then stabilize this swing to not overshoot when stabilizing the 2-link pendulum upwards in the goal position.

### I.1.3   Acrobot Environment

We can see in Figure 9, an illustration of the starting state Figure 9 (a) with a small perturbed random initial start. It is a 2-link pendulum with the individual joints actuated [Brockman et al., 2016]. The 2-link pole starts in the downward position Figure 9 (a) and the goal is to swing the 2-link pendulum upwards and then balance the pole(s) upright by applying torques about their fixed points. This environment has the state of $[\theta_1, \dot{\theta}_1, \theta_2, \dot{\theta}_2]$ and a corresponding observation of $[\sin(\theta_1), \cos(\theta_1), \dot{\theta}_1, \sin(\theta_2), \cos(\theta_2), \dot{\theta}_2]$. Here the Acrobot environment is fully actuated, as no method has been able to solve the underactuated balancing problem [Yildiz et al., 2021, Zhong et al., 2019].

### I.2   Cancer Environment

We implemented a Cancer environment, that uses a simulation of a Pharmacokinetic-Pharmacodynamic (PK-PD) (a bio-mathematical model that represents dose-response relationships)

model of lung cancer tumor growth [Geng et al., 2017] under continuous dosing treatment of chemotherapy and radiotherapy. Where the same underlying model has also been used by others [Lim, 2018, Bica et al., 2020, Seedat et al., 2022]. Here, the tumor volume $V$ at time $t$ after diagnosis is defined as

$$\frac{dV(t)}{dt} = \left( \underbrace{\rho \log \left( \frac{K}{V(t)} \right)}_{\text{Tumor growth}} - \underbrace{\beta_c C(t)}_{\text{Chemotherapy}} - \underbrace{(\alpha_r d(t) + \beta_r d(t)^2)}_{\text{Radiotherapy}} \right) V(t) \tag{54}$$

$$\frac{dC(t)}{dt} = -\frac{C(t)}{2} + c(t) \tag{55}$$

Where the state space is $[V(t) \in \mathbb{R}_+, C(t) \in \mathbb{R}_+]$, with the $V(t)$ being the cancer volume $\text{cm}^3$, and $C(t)$ being the chemotherapy concentration in the patient. Whereas the action space inputs are $[c(t) \in [0,5], d(t) \in [0,2]]$, the chemotherapy input dose $c(t)$ and radiotherapy dose $d(t)$—which are given by the policy. Specifically, $\rho, K, \alpha_r, \beta_r, \beta_c$ are effect parameters [Geng et al., 2017], and are defined as in Table 9.

Table 9: PK-PD model parameters for the Cancer environment.

| Model | Variable | Parameter | Value |
|---|---|---|---|
| Tumor growth | Growth parameter | $\rho$ | $1.45 \times 10^{-2}$ |
| | Carrying capacity | $K$ | 30.0 |
| Radiotherapy | Radio cell kill ($\alpha$) | $\alpha_r$ | 0.0398 |
| | Radio cell kill ($\beta$) | $\beta_r$ | Set s.t. $\alpha/\beta = 10$ |
| Chemotherapy | Chemo cell kill | $\beta_c$ | 0.028 |

As detailed above, the chemotherapy drug concentration state $C(t)$ follows an exponential decay relationship with a half-life of one day, and $c(t)$ represents a dose of Vinblastine up to the concentration of $5.0 mg/m^3$ per time in a day. Whereas the radiotherapy concentration $d(t)$ represents $2.0 G_y$ fractions of radiotherapy, where $G_y$ is the Gray ionizing radiation dose. Particularly, we set $\delta_a = .4$ days, and simulate each ODE system at a time resolution of $\delta_{\text{sim}} = 0.04$ days, up to an episode length of $T = 25$ days.

The goal state here is to reduce the cancer tumor volume to zero, alongside zero chemotherapy concentration in the patient, i.e. $q^* = [0, 0]$. Moreover, upon restarting the environment the initial state $s$ is sampled from the uniform distribution of $s_0 \sim \mathcal{U}[[1120, 0], [1138, 0]]$. Here we assume the patient starts with stage four cancer, the largest stage, with a tumor diameter of approximately 13cm diameter, assuming that the tumor is spherical. Furthermore, we use a larger growth parameter $\rho$ than the nominal one reported by Geng et al. [2017] to model aggressive cancer tumors—which is of high interest.

## J  Benchmark Method Implementation Details

All benchmark policies consist of an observing policy and an action policy. To ensure competitive benchmarks, each of the following benchmarks are specific ablations of our method, Active Observing Control. Therefore, for a given trained probabilistic dynamics model, the same action policy planner was used across all the benchmark methods, that of the same MPC MPPI planner with the hyperparameters fixed to the below values.

### J.1  Probabilistic Dynamics Model

To train our continuous-time and discrete-time probabilistic dynamics model we use a deep ensemble of fully connected multi-layer perceptions [Lakshminarayanan et al., 2017], adapted for multi-dimensional inputs [Chua et al., 2018]. Specifically, for each individual model in the ensemble, we use a 3-layer multilayer perceptron (MLP) of 256 units, with $\tanh$ activation functions. We also use the negative log-likelihood loss to train each model in the ensemble separately—training each model

for the same number of epochs with a different random seed, where the ensemble has $M = 5$ total models. We use Xavier weight initialization and output the log variance, following the setup by Chua et al. [2018]. All dynamics models are implemented in PyTorch [Paszke et al., 2019a], and trained with an Adam optimizer [Kingma and Ba, 2017] with a learning rate of 1e-4. For each environment we train two dynamics models on the offline dataset, that of a continuous-time probabilistic model using the time delta input $\delta$, and a discrete-time probabilistic model—which is the exact same model architecture without the additional time delta input $\delta$.

All dynamics models predict the next state difference, $\Delta z(t + \delta)$, and we construct the state as $z(t + \delta) = z(t) + \Delta z(t + \delta)$—following the standard suggestion of [Deisenroth and Rasmussen, 2011].

For training, using the whole collected dataset we pre-process this by a standardization step, to make each dimension of the samples have zero mean and unit variance (by taking away the mean for each dimension and then dividing by the standard deviation for each dimension)—we also use this step during run-time for each dynamics model. Furthermore, we train all the baseline models on all the samples collected in the offline dataset (all samples are training data) Lakshminarayanan et al. [2017]. Specifically, we train all the dynamics models on a given offline dataset by training each model until convergence, each for at least 12 hours.

## J.2   Observing Policies

We implement two discrete-time methods. First by learning a discrete-time uncertainty-aware dynamics model (an ablation of the exact same model and hyperparameters, without the time interval input $\delta$ to predict the next state for) on the same offline dataset $\mathcal{D}$. Second, we use this discrete-time uncertainty-aware dynamics model to create two benchmark methods, that of **Discrete Planning** that samples the state at each action time interval $\delta_a$ and **Discrete Monitoring** a discrete ablation of our observing policy (Algorithm 3)—that uses the same reward distribution estimation and determines the discrete action sequence to execute up until the reward uncertainty crosses the threshold $\tau$—at a discrete-time resolution of $\delta_a$.

Moreover, we also benchmark against **Continuous Planning** that uses our trained continuous-time uncertainty-aware dynamics model and takes a sample of the state at regular time intervals of a multiple of $\delta_a$. Finally, we also compare with a random action policy, **Random** that samples the state at each action interval $\delta_a$.

For each policy method, we tune the hyperparameter $\tau$ is one exists following the procedure outlined in Section 4.2. We then keep this fixed and constant throughout all run-time experiments for each policy method for each environment—unless explicitly noted that we modify it, as in Section 5.3. For all policies we limit the minimum $\Delta_{i+1}$ to be $\delta_a$ (as actions take time to execute and interact with the environment, therefore we want to avoid observing the same state again immediately, for fast-moving states), and the maximum $H$ the MPC planned action horizon.

## J.3   MPPI Implementation

We use the MPPI algorithm, with pseudocode and is further described in Appendix E. Specifically, as is recommended by Lutter et al. [2021] we optimized the MPPI hyperparameters through a grid search with the continuous-time probabilistic dynamics model for a single environment setting, that of the Cartpole environment, and fix these for planning with all the learned dynamics models throughout. Particularly, our final optimized hyperparameter combination is $N = 20, M = 1,000, \lambda = 0.01, \sigma = 1.0$. Where $\Sigma$ the MPPI action noise is defined as:

$$\Sigma = \begin{cases} [\sigma^2] & \text{if } d_\mathcal{A} = 1 \\ \begin{bmatrix} \sigma^2 & 0.5\sigma^2 \\ 0.5\sigma^2 & \sigma^2 \end{bmatrix} & \text{if } d_\mathcal{A} = 2 \end{cases} \tag{56}$$

Where the Cartpole and Pendulum environments have $d_\mathcal{A} = 1$ and Acrobot and Cancer environments has $d_\mathcal{A} = 2$. These hyperparameters were found by searching over a grid of possible values, which is detailed in Table 10.

Table 10: MPPI hyperparameter grid search sweep values.

| Hyperparameter | Grid values searched over |
|---|---|
| $K$ | $\{1, 2, 4, 8, 16, 20, 40, 50, 60, 70, 80, 90, 100, 128, 256, 512, 1024, 2048\}$ |
| $G$ | $\{1, 2, 4, 8, 16, 20, 40, 50, 60, 70, 80, 90, 100, 128, 256, 500, 1000, 2000, 4000, 8000\}$ |
| $\lambda$ | $\{0.00001, 0.0001, 0.001, 0.01, 0.1, 0.5, 0.8, 1.0, 1.5, 2.0, 10.0, 100.0, 1000.0\}$ |
| $\sigma$ | $\{0.00001, 0.0001, 0.001, 0.01, 0.1, 0.5, 0.8, 1.0, 1.5, 2.0, 10.0, 100.0, 1000.0\}$ |

# K  Evaluation Metrics

An uncertainty-aware dynamics model is trained for each environment on an offline dataset $\mathcal{D}$ of collected trajectories consisting of 1e6 samples. The trained dynamics model weights are then frozen when used at run-time. We record the average undiscounted utility $\mathcal{U}$ (Equation (1)), average undiscounted reward $\mathcal{R}$ and the number of observations taken $\mathcal{O}$, after running one episode of the benchmark method—and repeat this for 100 random seed runs for each result with their 95% confidence intervals throughout. We use a fixed observation cost of $c = 50$ throughout. Moreover, we normalize $\mathcal{R}$ and $\mathcal{U}$ following the standard normalization of offline-RL [Yu et al., 2020]—normalized to the interval of 0 to 100, where a score of 0 corresponds to a random policy performance, and 100 to an expert—which we assume here is the Continuous Planning benchmark. Furthermore, we also track the metric of total planning time taken to plan the next action and time to schedule a sample and perform all experiments using a single Intel Core i9-12900K CPU @ 3.20GHz, 64GB RAM with a Nvidia RTX3090 GPU 24GB.

# L  Additional Experiments

## L.1  Ablation of Thresholding the State Uncertainty

Intuitively we threshold the variance of the value function rather than the variance of the state as it is possible to achieve a task with a very certain value despite having uncertain states. For example, when there exist multiple ways to achieve a task—and we know that multiple action trajectories guarantee this where we can take any; however, we are uncertain about which one.

We performed an additional ablation experiment where we threshold the variance of state instead and compare the utility against Active Observing Control that thresholds the variance of the predicted reward instead.

Table 11: Ablation experiment of using a threshold on the variance of state instead of the predicted reward, for the Cancer environment.

| Policy | Utility $\mathcal{U}$ |
|---|---|
| Active Observing Control (Reward Uncertainty Threshold) | 106±0.789 |
| Active Observing Control (State Uncertainty Threshold) | 103±1.6 |

For each policy, we tuned the threshold following the tuning setup, in Section 4.2. Here we evaluated the utility metric over 10 random seeds for each policy on the Cancer environment. As tabulated in Table 11, it is possible to threshold on the state variance for an acceptable performing policy, however, it is preferable and more intuitive to threshold the variance of the predicted reward uncertainty.

Furthermore, we performed an additional ablation experiment where we threshold the variance of the state instead and compare this to the utility of Active Observing Control that thresholds the variance of the predicted reward instead, on the Cartpole environment, as tabulated in Table 12.

We again tuned each observing policy's threshold following the tuning setup in Section 4.2. Here we evaluated the utility metric over 100 random seeds. Empirically we observe that is indeed possible to threshold the state variance for an acceptable performing observing policy. However, it is again preferable and more intuitive to threshold the variance of the predicted reward uncertainty.

We also highlight the Cartpole environment is notably more complicated than the Cancer environment, as it is an underactuated system, as it has two degrees of freedom $[x, \theta]$, however only the carts

Table 12: Ablation experiment of using a threshold on the variance of state instead of the predicted reward, for the Cartpole environment.

| Policy | Utility $\mathcal{U}$ |
|---|---|
| Active Observing Control (Reward Uncertainty Threshold) | **144±5.51** |
| Active Observing Control (State Uncertainty Threshold) | 98.4±2.64 |

position is actuated, leaving $\theta$ indirectly controlled. We note that the control actuator is bounded and limited, and the force is such that the Cartpole cannot be directly swung up—rather it must gain momentum through a swing and then stabilize this swing to not overshoot when stabilizing the pole upwards in the goal position, as indicated when the tip of the pole reaches the center of the red target square.

## L.2  Sparse Reward Environment

To investigate how well Active Observing Control generalizes to other environments that have possibly sparse reward functions, we performed an additional sparse reward experiment. We implemented a sparse Pendulum environment [Chakraborty et al., 2022] which only provides a reward when the Pendulum state angle $\theta$ is within $\pm\theta_{\text{CutOff}}$ degrees from the goal state upright ($\theta^* = 0$). That is using a sparse reward function defined by $r'(s, a, t) = r(s, a, t) \cdot \mathbf{1}\{|\theta| < \theta_{\text{CutOff}}\}$. This allows us to vary the degree of sparsity by decreasing $\theta_{\text{CutOff}}$ to create a sparser reward for the Pendulum environment. This is tabulated in Table 13, with each result averaged over 100 random seeds, and we normalize scores to 100 for Continuous Planning with $\theta_{\text{CutOff}} = 180°$.

Table 13: Sparse reward Pendulum environment.

| $\theta_{\text{CutOff}}$ | Policy | Utility $\mathcal{U}$ | Reward $\mathcal{R}$ | Observations $\mathcal{O}$ |
|---|---|---|---|---|
| $\theta_{\text{CutOff}} = 180°$ | Random | 0±0 | 0±0 | 13±0 |
| $\theta_{\text{CutOff}} = 180°$ | Continuous Planning | 100±9.48 | 100±9.48 | 50±0 |
| $\theta_{\text{CutOff}} = 60°$ | Continuous Planning | 18.8±17.7 | 18.8±17.7 | 50±0 |
| $\theta_{\text{CutOff}} = 5°$ | Continuous Planning | -7.81±0.00125 | -7.81±0.00125 | 50±0 |
| $\theta_{\text{CutOff}} = 180°$ | **Active Observing Control** | 109±13.8 | 98.5±10.2 | 48±0.954 |
| $\theta_{\text{CutOff}} = 60°$ | **Active Observing Control** | 86.7±13.5 | 12±16.5 | 35.4±4.08 |
| $\theta_{\text{CutOff}} = 5°$ | **Active Observing Control** | 223±2.32 | -7.81±0.00146 | 4.8±0.452 |

We note that Active Observing Control relies on MPC being feasible in the environment and that it is possible to solve the environment with the chosen MPC decision policy $\pi$ with regular observing at each time step. In our experimental setup, this translates that the baseline of Continuous Planning should be able to solve the environment to achieve an optimal reward (i.e., $\mathcal{R} = 100$).

The above results demonstrate the well-known [Karnchanachari et al., 2020] limitation of MPC in general, that a vanilla MPC decision policy $\pi$ struggles in environments with sparse rewards. We highlight that this is not unique to MPC, and standard RL methods in general significantly struggle with sparse rewards in continuous-state and continuous-action tasks [Chakraborty et al., 2022] without additional assumptions or data (reward shaping or using expert demonstrations). This arises due to the limited receding planning horizon time $H$, however, there exists works to combine trajectory optimization (MPC) with learning a reward estimation [Karnchanachari et al., 2020, Hoeller et al., 2020]. This is an interesting direction for future work, though we believe this is out of scope for this paper.

## L.3  Increasing the Planning Resolution of MPC (decreasing $\delta_a$)

We note that planning with a higher resolution MPC planner (smaller $\delta_a$) for all the baselines will make the baselines perform on par with Active Observing Control for that $\delta_a$. However, our motivation is to compare the observing policies $\rho$ of each baseline, therefore in all the implemented baselines we use the same decision policy $\pi$ of the same MPC algorithm with the exact same hyperparameters.

We performed an additional re-run of the baselines on the Cancer environment, with varying $\delta_a = \{0.1, 0.05, 0.02, 0.01\}$ and each result averaged over 100 random seeds, as tabulated in Table 14. We

kept the MPC planning horizon time $H$ fixed, to the same value as in our original experiments, and let the number of planning time steps $K$ vary, i.e., $K = \lceil \frac{H}{\delta_a} \rceil$.

Table 14: Increasing the planning resolution of MPC (decreasing $\delta_a$) for the Cancer environment.

| Discretization interval $\delta_a$ | Policy | Utility $\mathcal{U}$ | Reward $\mathcal{R}$ | Observations $\mathcal{O}$ |
|---|---|---|---|---|
| 0.1 | Random | 0±0 | 0±0 | 13±0 |
| 0.1 | Discrete Planning | 92.2±1.11 | 92.2±1.11 | 13±0 |
| 0.1 | Discrete Monitoring | 91.1±1.72 | 85.9±1.7 | 4.94±0.109 |
| 0.1 | Continuous Planning | 100±0.597 | 100±0.597 | 13±0 |
| 0.1 | **Active Observing Control** | **105±0.75** | 98.4±0.707 | 3.42±0.102 |
| 0.05 | Random | 0±0 | 0±0 | 13±0 |
| 0.05 | Discrete Planning | 96±1.54 | 95.6±1.68 | 13±0 |
| 0.05 | Discrete Monitoring | 96.3±1.39 | 91.6±1.47 | 7.18±0.153 |
| 0.05 | Continuous Planning | 100±1.36 | 100±1.48 | 13±0 |
| 0.05 | **Active Observing Control** | **104±1.29** | 98.1±1.33 | 4.95±0.118 |
| 0.02 | Random | 0±0 | 0±0 | 13±0 |
| 0.02 | Discrete Planning | 95±1.25 | 92.8±1.81 | 13±0 |
| 0.02 | Discrete Monitoring | 95.9±1.17 | 88.2±1.7 | 6.5±0.196 |
| 0.02 | Continuous Planning | 100±1.17 | 100±1.7 | 13±0 |
| 0.02 | **Active Observing Control** | **98.6±0.967** | 88.8±1.43 | 2.77±0.0839 |
| 0.01 | Random | 0±0 | 0±0 | 13±0 |
| 0.01 | Discrete Planning | 102±0.806 | 105±1.87 | 13±0 |
| 0.01 | Discrete Monitoring | 102±0.844 | 98±1.93 | 5.89±0.0741 |
| 0.01 | Continuous Planning | 100±1.03 | 100±2.39 | 13±0 |
| 0.01 | **Active Observing Control** | **103±0.937** | 97.8±2.06 | 4.79±0.117 |

Where for each $\delta_a$ we normalize utility to be between 100 and 0, where 100 corresponds to Continuous planning at that $\delta_a$ and 0 a random policy. We note that $\delta_a = 0.1$ was used, as this is the default value for the ODE-RL environments [Yildiz et al., 2021].

Moreover, in light of the empirical results (Table 14), we can also ask what the performance looks like if we only normalize utility to Continuous planning with $\delta_a = 0.1$ (the best possible reward) for each baseline whilst varying $\delta_a$, as tabulated in Table 15.

Interestingly we observe all baselines degrading in performance, however, AOC still has the highest utility amongst the baselines for a set $\delta_a$. Intuitively the MPC MPPI algorithm produces less optimal action trajectory $a$ plans when the number of discrete actions $K$ in the discrete action sequence increases for a fixed number of parallel rollouts $G$. We note that this is inherent to using an MPC planner [Williams et al., 2017], and could be improved by increasing $G$ for decreasing $\delta_a$, however, that is out of scope for this work.

### L.4 Discrete Time Controller Trained on an Equidistant Dataset

The Continuous Planning baseline (which uses a continuous-time dynamics model) can perform better than the Discrete Planning baseline (which uses a discrete-time dynamics model), as it has a more accurate dynamics model—when both are trained on the irregularly sampled offline dataset $\mathcal{D} = \{(z(t_i), a(t_i), \Delta_i)\}_{i=1}^N$. As the offline dataset of state-action trajectories $\mathcal{D}$, $\Delta_{i+1}$ has irregular times between state-action samples $\Delta_{i+1} \sim \text{Exp}(\bar{\Delta})$, with a mean of $\bar{\Delta} = \delta_a$ seconds.

We test what would happen if the discrete-time dynamics model (for the Discrete Planning and Discrete Monitoring baselines) was instead trained on a regular (equidistant) sampled offline dataset $\mathcal{D}$, i.e., $\Delta_{i+1} = \bar{\Delta} = \delta_a$, we performed an additional experiment. This is in the Cancer environment, with each result averaged over 100 random seeds, as Tabulated in Table 16.

Empirically this agrees with our intuition that the discrete-time dynamics model baselines do indeed perform better when trained on an offline dataset that is regularly sampled. However, the discrete-time baselines still underperform Active Observing Control, due to their inherent time discretization error, Figure 6.

Table 15: Increasing the planning resolution of MPC (decreasing $\delta_a$) for the Cancer environment. Here we only normalize utility to Continuous planning with $\delta_a = 0.1$ (the best possible reward) for each baseline whilst varying $\delta_a$.

| Discretization interval $\delta_a$ | Policy | Utility $\mathcal{U}$ | Reward $\mathcal{R}$ | Observations $\mathcal{O}$ |
|---|---|---|---|---|
| 0.1 | Random | 0±0 | 0±0 | 13±0 |
| 0.1 | Discrete Planning | 92.2±1.11 | 92.2±1.11 | 13±0 |
| 0.1 | Discrete Monitoring | 91.1±1.72 | 85.9±1.7 | 4.94±0.109 |
| 0.1 | Continuous Planning | 100±0.597 | 100±0.597 | 13±0 |
| 0.1 | **Active Observing Control** | **105±0.75** | 98.4±0.707 | 3.42±0.102 |
| 0.05 | Random | 0±0 | 0±0 | 13±0 |
| 0.05 | Discrete Planning | 85.6±1.44 | 85.6±1.44 | 13±0 |
| 0.05 | Discrete Monitoring | 85.9±1.3 | 82.1±1.26 | 7.18±0.153 |
| 0.05 | Continuous Planning | 89.3±1.27 | 89.3±1.27 | 13±0 |
| 0.05 | **Active Observing Control** | **92.9±1.2** | 87.7±1.14 | 4.95±0.118 |
| 0.02 | Random | 0±0 | 0±0 | 13±0 |
| 0.02 | Discrete Planning | 71.8±1.3 | 71.8±1.3 | 13±0 |
| 0.02 | Discrete Monitoring | 72.8±1.22 | 68.6±1.23 | 6.5±0.196 |
| 0.02 | Continuous Planning | 77.1±1.22 | 77.1±1.22 | 13±0 |
| 0.02 | **Active Observing Control** | **75.6±1.01** | 69±1.03 | 2.77±0.0839 |
| 0.01 | Random | 0±0 | 0±0 | 13±0 |
| 0.01 | Discrete Planning | 63.2±1.03 | 63.2±1.03 | 13±0 |
| 0.01 | Discrete Monitoring | 63.8±1.07 | 59.2±1.06 | 5.89±0.0741 |
| 0.01 | Continuous Planning | 60.3±1.32 | 60.3±1.32 | 13±0 |
| 0.01 | **Active Observing Control** | **64.4±1.19** | 59.1±1.13 | 4.79±0.117 |

Table 16: Results for the Cancer environment where the discrete-time dynamics model is trained on a regular (equidistant) sampled offline dataset $\mathcal{D}$.

| $\Delta_{i+1}$ Distribution | Policy | Utility $\mathcal{U}$ | Reward $\mathcal{R}$ | Observations $\mathcal{O}$ |
|---|---|---|---|---|
| Irregularly sampled, $\Delta_{i+1} \sim \text{Exp}(\bar{\Delta})$ | Random | 0±0 | 0±0 | 13±0 |
| Regularly sampled, $\Delta_{i+1} = \bar{\Delta}$ | Discrete Planning | 95.5±0.667 | 95.5±0.667 | 13±0 |
| Regularly sampled, $\Delta_{i+1} = \bar{\Delta}$ | Discrete Monitoring | 96.1±1.83 | 90.1±1.83 | 3.75±0.127 |
| Irregularly sampled, $\Delta_{i+1} \sim \text{Exp}(\bar{\Delta})$ | Continuous Planning | 100±0.594 | 100±0.594 | 13±0 |
| Irregularly sampled, $\Delta_{i+1} \sim \text{Exp}(\bar{\Delta})$ | **Active Observing Control** | **105±0.755** | 98.5±0.713 | 3.4±0.102 |

## L.5 Sensitivity of AOC to $\delta_a$

$\delta_a$ is often chosen for a user, as it is the average time interval between two state-action points in a trajectory in the collected offline dataset $\mathcal{D}$.

Here MPC depends on a given $\delta_a$, however, the observing policy is orthogonal to $\delta_a$. As outlined in Appendix L.3, reducing $\delta_a$ whilst keeping the planning time horizon $H$ and the number of parallel rollouts $G$ the same, reduces the MPC MPPI action trajectory quality, leading to control performance that achieves a lower reward, and hence utility.

We performed an additional experiment to verify this on the Cancer environment, varying $\delta_a = \{0.1, 0.05, 0.02, 0.01\}$, each result averaged over 100 random seeds, as tabulated in Table 17. Here we normalize utility to Continuous planning with $\delta_a = 0.1$ (the best possible reward) for each baseline whilst varying $\delta_a$.

Table 17: Results for the Cancer environment where vary $\delta_a$ for AOC.

| Discretization interval $\delta_a$ | Policy | Utility $\mathcal{U}$ | Reward $\mathcal{R}$ | Observations $\mathcal{O}$ |
|---|---|---|---|---|
| 0.1 | Random | 0±0 | 0±0 | 13±0 |
| 0.1 | Continuous Planning | 100±0.597 | 100±0.597 | 13±0 |
| 0.1 | **Active Observing Control** | **105±0.75** | 98.4±0.707 | 3.42±0.102 |
| 0.05 | **Active Observing Control** | 92.9±1.2 | 87.7±1.14 | 4.95±0.118 |
| 0.02 | **Active Observing Control** | 75.6±1.01 | 69±1.03 | 2.77±0.0839 |
| 0.01 | **Active Observing Control** | 64.4±1.19 | 59.1±1.13 | 4.79±0.117 |

## L.6 Evaluation of Reward Uncertainty

The reliability of the reward uncertainty depends on the reliability of the probabilistic dynamics model $\hat{f}_\theta$ to provide a good predictive uncertainty for a future state. We highlight that the predictive uncertainty of the dynamics model depends only on the training, that of training on the offline dataset of trajectories $\mathcal{D}$—which is only performed once. At run-time the dynamics model is only used for inference, therefore its predictive uncertainty for the state-action space will remain the same, independent of the number of samples of the state taken in a given evaluation episode.

To illustrate this point, we performed an additional experiment on the Cancer environment where, at run-time, we vary the number of observations (samples of the state taken) in an episode and evaluate the negative log-likelihood (NLL) [Lakshminarayanan et al., 2017] of the predicted reward uncertainty to the ground truth reward uncertainty for 100 random seeds, as tabulated in Table 18.

Table 18: Evaluation of reward uncertainty of AOC on the Cancer environment.

| State Observations | NLL |
|---|---|
| 1 | $8.261 \pm 1.124$ |
| 10 | $8.138 \pm 1.060$ |
| 100 | $8.190 \pm 1.052$ |
| 1,000 | $8.241 \pm 0.973$ |
| 10,000 | $7.933 \pm 0.288$ |
| 100,000 | $7.988 \pm 0.093$ |

## L.7 Relation Between Rewards $\mathcal{R}$ and the Fixed Observation Cost $c$

An exact theoretical relation between rewards $\mathcal{R}$ and the fixed observation cost $c$ is *intractable* to provide for the general case.

We encounter this intractability even for our simple analytic system used to prove Proposition 2.1. As let us start with Equation (1), $\mathcal{U} = \int_0^T r(s(t), a(t), t)\mathrm{d}t - c|\{t_i : t_i \in [0, T]\}|$, where the Reward is $\mathcal{R} = \int_0^T r(s(t), a(t), t)\mathrm{d}t$ and Cost is $\mathcal{C} = c|\{t_i : t_i \in [0, T]\}|$. For even a simple regular observing policy (Appendix C), $\mathbb{E}_{\rho^{(\text{reg},\delta)}}[\mathcal{U}]$, which is a function of $\delta \in (0, T)$, as well as the reward function $r(s(t), a(t), t)$ and the observation cost $c$.

If we keep $r$, $\pi$, $c$ fixed, and attempt to find $\rho^* = \mathrm{argmax}_\rho \mathbb{E}[\mathcal{U}]$ by only varying $\delta$, that is by $\frac{\mathrm{d}}{\mathrm{d}\delta}\mathbb{E}[\mathcal{U}] = 0$, calculating a closed-form solution for $\frac{\mathrm{d}}{\mathrm{d}\delta}\mathbb{E}[\mathcal{U}]$ is intractable.

Instead, we can empirically simulate different values of $\delta$, then see which grid sweep value of $\delta$ maximizes $\mathbb{E}[\mathcal{U}]$, such that approximately $\delta^* = \mathrm{argmax}_\delta \mathbb{E}[\mathcal{U}]$, this value can then be used to calculate the Reward $\mathcal{R} = \int_0^T r(s(t), a(t), t)\mathrm{d}t$.

We do this below on the Cancer environment where we use a regular observing policy, that of Continuous Planning, and vary $\delta$ over a grid sweep such that different run settings of the policy take different observations, as tabulated in Table 19. We start with setting $c = 1$ and only normalize the Reward $\mathcal{R}$ to 100 for the most frequent samples possible ($\mathcal{O} = 13$), each result calculated for 100 random seeds.

Table 19: Cancer environment where we use a regular observing policy, that of Continuous Planning, and vary $\delta$ over a grid sweep such that different run settings of the policy take different observations.

| Observations $\mathcal{O}$ | Reward $\mathcal{R}$ | Utility $\mathcal{U}$ |
|---|---|---|
| $13 \pm 0$ | $100 \pm 0.616$ | 87 |
| $7 \pm 0$ | $99.3 \pm 0.656$ | **92.3** |
| $4 \pm 0$ | $96 \pm 0.821$ | 92 |
| $2 \pm 0$ | $90.7 \pm 1.01$ | 88.7 |
| $1 \pm 0$ | $89.2 \pm 1.1$ | 88.2 |

For a given cost $c$, i.e., in this case $c = 1$, we see that there exists an approximate number of observations that maximize the Utility, here for $c = 1$ this is $\mathcal{O} = 7$. This allows us to vary the observation cost $c$ numerically, and for each observation cost, determine the number of observations (or regular observing frequency) that maximizes the Utility $\mathcal{U}$ ($\mathcal{O}^* = \text{argmax}_{\mathcal{O}}\mathbb{E}[\mathcal{U}]$) and determine what the Reward $\mathcal{R}$ for this $\mathcal{O}^*$ is. This is tabulated in Table 20.

Table 20: Cancer environment where we vary the observation cost $c$ numerically, and for each observation cost, determine the number of observations (or regular observing frequency) that maximizes the Utility $\mathcal{U}$ ($\mathcal{O}^* = \text{argmax}_{\mathcal{O}}\mathbb{E}[\mathcal{U}]$) and determine what the Reward $\mathcal{R}$ for this $\mathcal{O}^*$ is.

| Fixed Observation Cost $c$ | Optimal Utility $\mathcal{U}^*$ | Optimal number of Observations $\mathcal{O}^*$ | Reward $\mathcal{R}$ |
|---|---|---|---|
| 0 | 100 | 13 | 100 |
| 1 | 92.3 | 7 | 99.3±0.656 |
| 2 | 88 | 4 | 96±0.821 |
| 3 | 86.2 | 1 | 89.2±1.1 |

Empirically we observe the relation between the $c$ and $\mathcal{R}$ is that the fixed observation cost $c$ is negatively correlated to the Reward $\mathcal{R}$, when using a policy that maximizes the Utility $\mathcal{U}$, Equation (1).

## L.8  Dependence on the Accuracy of the Learned Dynamics Model

To understand the dependence on the accuracy of the learned dynamics model further, we performed an additional experiment ablation, where we benchmarked all methods with a less accurate dynamics model—training all the dynamics models with fewer samples (10% of the total amount of samples used in training the dynamics models presented in the main paper), here trained on 100,000 samples. This is tabulated in Table 21.

We observe that AOC still achieves a high average utility $\mathcal{U}$ on all environments, outperforming the competing Continuous Planning and Discrete Monitoring methods. Specifically, this empirically further verifies the key theoretical contribution that regular observing is not optimal and that irregularly observing can achieve a higher expected utility.

Table 21: Ablation with a less accurate dynamics model—training all the dynamics models with fewer samples (10% of the total amount of samples used in training the dynamics presented in the main paper), here trained on 100,000 samples. We observe that AOC still achieves a high average utility $\mathcal{U}$ on all environments, outperforming the competing Continuous Planning and Discrete Monitoring methods. Normalized utilities $\mathcal{U}$, rewards $\mathcal{R}$ and observations $\mathcal{O}$ for the benchmark methods, across each environment. Results are averaged over 1,000 random seeds, with ± indicating 95% confidence intervals. Utilities and rewards are undiscounted and normalized to be between 0 and 100, where 0 corresponds to a Random agent, and 100 corresponds to the expert, that of Continuous Planning, taking state observations at every $\delta_a$.

| Policy | Cancer | | | Acrobot | | | Cartpole | | | Pendulum | | |
|---|---|---|---|---|---|---|---|---|---|---|---|---|
| | $\mathcal{U}$ | $\mathcal{R}$ | $\mathcal{O}$ | $\mathcal{U}$ | $\mathcal{R}$ | $\mathcal{O}$ | $\mathcal{U}$ | $\mathcal{R}$ | $\mathcal{O}$ | $\mathcal{U}$ | $\mathcal{R}$ | $\mathcal{O}$ |
| Random | 0±0 | 0±0 | 13±0 | 0±0 | 0±0 | 50±0 | 0±0 | 0±0 | 50±0 | 0±0 | 0±0 | 50±0 |
| Discrete Planning | 99.3±0.701 | 99.3±0.701 | 13±0 | 67.2±6.46 | 67.2±6.46 | 50±0 | 117±9.05 | 117±9.05 | 50±0 | 51±3.13 | 51±3.13 | 50±0 |
| Discrete Monitoring | 74.4±2.76 | 69.1±2.73 | 4.51±0.161 | 142±6.4 | 40±6.37 | 5.48±0.111 | 709±36.9 | -97.1±36.9 | 6±0 | 249±1.18 | 13.6±1.94 | 6.3±0.24 |
| Continuous Planning | 100±0.754 | 100±0.754 | 13±0 | 100±6.74 | 100±6.74 | 50±0 | 100±2.51 | 100±2.51 | 50±0 | 100±2.87 | 100±2.87 | 50±0 |
| **Active Observing Control** | **104±0.767** | 98.3±0.735 | 3.62±0.0968 | **157±6.93** | 54.5±6.97 | 5.29±0.0948 | **825±2.35** | 0.427±2.35 | 5±0 | **277±1.28** | 39.4±1.25 | 5.99±0.0527 |

## L.9  Extending AOC to Work with a Learned Reward Model

To investigate whether AOC can also be used with a learned reward model, we performed an additional experiment, where we trained an MLP reward model (4-layer MLP with 128 units and Tanh activations) from the offline dataset in all the benchmarks—this is tabulated in Table 22.

We observe that AOC still achieves a high average utility $\mathcal{U}$ on the Cancer environment, outperforming the competing Continuous Planning and Discrete Monitoring methods. This empirically verifies that our proposed initial approach can still perform well using a learned reward model and that the theoretical contribution still holds.

Table 22: We use a learned MLP reward model (4-layer MLP with 128 units and Tanh activations) from the offline dataset in all the benchmarks. We observe that AOC still achieves a high average utility $\mathcal{U}$ on the Cancer environment, outperforming the competing Continuous Planning and Discrete Monitoring methods. Normalized utilities $\mathcal{U}$, rewards $\mathcal{R}$ and observations $\mathcal{O}$ for the benchmark methods, across the Cancer environment. Results are averaged over 1,000 random seeds, with $\pm$ indicating 95% confidence intervals. Utilities and rewards are undiscounted and normalized to be between 0 and 100, where 0 corresponds to a Random agent, and 100 corresponds to the expert, that of Continuous Planning, taking state observations at every $\delta_a$.

| Policy | $\mathcal{U}$ | $\mathcal{R}$ | $\mathcal{O}$ |
|---|---|---|---|
| Random | 0±0 | 0±0 | 13±0 |
| Discrete Planning | 91.4±0.383 | 91.4±0.383 | 13±0 |
| Discrete Monitoring | 95.1±0.4 | 91.6±0.392 | 7.63±0.0631 |
| Continuous Planning | 100±0.151 | 100±0.151 | 13±0 |
| **Active Observing Control** | **105±0.178** | 98.9±0.168 | 3.38±0.0303 |

## L.10 AOC Also Empirically Works for Non-linear State Transformations

We performed an additional experiment to investigate AOC's performance within environments that utilize observations stemming from non-linear state transformations. We tailored the existing Cancer environment to render observations via the non-linear state transformation function $z(t) = 0.1(s(t) + \epsilon(t))^2 + (s(t) + \epsilon(t))$. The subsequent results, conducted across 1,000 random seeds, are outlined in Table 23. We observe that AOC still achieves a high average utility $\mathcal{U}$ on all environments, outperforming the competing Continuous Planning and Discrete Monitoring methods. Specifically, this empirically further verifies our key theoretical contribution that regular observing is not optimal and that irregularly observing can achieve a higher expected utility.

Table 23: We adapted the existing Cancer environment to render observations via the non-linear state transformation function $z(t) = 0.1(s(t) + \epsilon(t))^2 + (s(t) + \epsilon(t))$. We observe that AOC still achieves a high average utility $\mathcal{U}$ on all environments, outperforming the competing Continuous Planning and Discrete Monitoring methods. Normalized utilities $\mathcal{U}$, rewards $\mathcal{R}$ and observations $\mathcal{O}$ for the benchmark methods, across the Cancer environment. Results are averaged over 1,000 random seeds, with $\pm$ indicating 95% confidence intervals. Utilities and rewards are undiscounted and normalized to be between 0 and 100, where 0 corresponds to a Random agent, and 100 corresponds to the expert, that of Continuous Planning, taking state observations at every $\delta_a$.

| Policy | $\mathcal{U}$ | $\mathcal{R}$ | $\mathcal{O}$ |
|---|---|---|---|
| Random | 0±0 | 0±0 | 13±0 |
| Discrete Planning | 91.7±0.397 | 91.7±0.397 | 13±0 |
| Discrete Monitoring | 90.5±0.559 | 85.4±0.548 | 5.25±0.0329 |
| Continuous Planning | 100±0.21 | 100±0.21 | 13±0 |
| **Active Observing Control** | **104±0.221** | 98.9±0.208 | 4.68±0.0315 |

## L.11 Further Empirical Validation in Other Real-World Scenarios

To verify the theoretical contribution in additional real-world scenarios, we show empirically in the following that AOC can successfully operate in the real-world environments (medical and engineering) of an accurate *Glucose environment* (controlling the injected insulin for a diabetic patient [Lenhart and Workman, 2007]), a (Human Immunodeficiency Virus) *HIV environment* (controlling the chemotherapy dose for affecting the infectivity of HIV in a patient [Butler et al., 1997]), and a *Quadrotor environment* (controlling the actuators of an unmanned aerial vehicle [Nonami et al., 2010]). Each environment represents a unique challenge and has direct implications in medical and engineering applications, thus reflecting the broad applicability of our method. These are:

- *Glucose environment*. Controlling the injected insulin for a diabetic patient to regulate their blood glucose level—here observations are costly as a blood test must be performed to measure the glucose level [Lenhart and Workman, 2007].

- *HIV environment.* Controlling the chemotherapy dose for affecting the infectivity of HIV in a patient—where observations are costly as a blood test must be performed to measure CD4*T cell levels [Butler et al., 1997].
- *Quadrotor environment.* Controlling the actuators of an unmanned aerial vehicle—where observations can be costly due to performing an expensive (power and compute) localization measure [Nonami et al., 2010].

We observe that AOC still achieves a high average utility $\mathcal{U}$ on all environments, outperforming the competing Continuous Planning and Discrete Monitoring methods, reinforcing our theoretical claims, and extending our empirical validation—as tabulated in Table 24.

Table 24: Normalized utilities $\mathcal{U}$, rewards $\mathcal{R}$, and observations $\mathcal{O}$ for the benchmark methods, across each environment. AOC performs the best across all environments. Results are averaged over 1,000 random seeds, with $\pm$ indicating 95% confidence intervals. Utilities and rewards are undiscounted and normalized to be between 0 and 100, where 0 corresponds to a Random agent, and 100 corresponds to the expert, that of Continuous Planning, taking state observations at every $\delta_a$.

| Policy | Glucose | | | HIV | | | Quadcoptor | | |
| | $\mathcal{U}$ | $\mathcal{R}$ | $\mathcal{O}$ | $\mathcal{U}$ | $\mathcal{R}$ | $\mathcal{O}$ | $\mathcal{U}$ | $\mathcal{R}$ | $\mathcal{O}$ |
|---|---|---|---|---|---|---|---|---|---|
| Random | 0±0 | 0±0 | 50±0 | 0±0 | 0±0 | 50±0 | 0±0 | 0±0 | 50±0 |
| Discrete Planning | 96±0.485 | 96±0.485 | 50±0 | 152±0.121 | 152±0.121 | 50±0 | 101±0.00882 | 101±0.00882 | 50±0 |
| Discrete Monitoring | 120±0.489 | 92.9±0.493 | 15.7±0.0466 | 2.77e+03±0.455 | 126±0.455 | 7±0 | 1.79e+03±0.199 | 101±0.0142 | 5.99±0.00518 |
| Continuous Planning | 100±0.39 | 100±0.39 | 50±0 | 100±0.431 | 100±0.431 | 50±0 | 100±0.015 | 100±0.015 | 50±0 |
| **Active Observing Control** | **126±0.41** | 99.9±0.39 | 17.5±0.0336 | **2.83e+03±1.56** | 107±0.878 | 5.75±0.0269 | **1.83e+03±0.0789** | 97.9±0.0258 | 5±0.00196 |

