# OpenReview forum: "Active Observing in Continuous-time Control"
_NeurIPS.cc/2023/Conference — NeurIPS 2023 poster_

### Official Review · Reviewer_mUFv · 2023-07-05

**Soundness:** 3 good
**Presentation:** 3 good
**Contribution:** 2 fair
**Rating:** 6
**Confidence:** 3

**Summary:**

This paper tackles the issue of determining the optimal timing for observations in continuous-time control with costly observations.
The authors formulate the problem and develop theoretical findings to provide insights about why observing at irregular intervals is advantageous in certain environments.
They then introduce a novel method called Active Observing Control, which involves taking observations at irregular intervals and performing continuous-time control. The adaptive time interval is achieved using a heuristic that compares the reward variance of the rollout with a threshold, leveraging a model learned from an offline dataset.

The effectiveness of this method is validated through experimental results, demonstrating that it outperforms alternative approaches that either utilize a discrete-time model for planning or employ a continuous-time model with regular observation intervals. Ablation studies illustrate how the proposed method can mitigate the discretization error associated with discrete-time formulation and show that the performance is not sensitive to the choice of the threshold.


**Strengths:**

- The proposed approach is novel.
- It provides insights from both theoretical and experimental results.
- It includes comprehensive implementation details and ablation results.


**Weaknesses:**

1. The proposed method appears to be heavily reliant on the accuracy of the learned model. This could limit the generalizability and robustness of the approach.
Specifically, the time interval is directly impacted by the uncertainty of the reward. Knowing the true reward function simplifies the problem to estimating uncertainty of the dynamics.
It is not clear whether the approach would work well in the general case, where the reward function is not known and need to be learned.

2. The following related works are not included:

   [1] Huang, Y. and Zhu, Q., 2020. Infinite-horizon linear-quadratic-gaussian control with costly measurements. arXiv preprint arXiv:2012.14925.

   [2]  Zhang, Z., Kirschner, J., Zhang, J., Zanini, F., Ayoub, A., Dehghan, M. and Schuurmans, D., 2022. Managing Temporal Resolution in Continuous Value Estimation: A Fundamental Trade-off. arXiv preprint arXiv:2212.08949.

   In particular, [1] adds the cost of observations to the original cost function based on the state and action, similar to this work, but in a discrete-time LQG setting.
   [2] studies the continuous-time stochastic LQR setting, where they formulate the costly measurement differently through the data budget.

3. The paper claims that the formulation is novel. However, it appears that Reference [1] above proposes a similar objective function, albeit within the Linear Quadratic Gaussian (LQG) setting. It would be beneficial if the authors could provide clarifications on the differences between their proposed formulation and that found in Reference [1].

4. I may be missing something but the method might not be truly continuos-time. The time resolution seems to be lower-bounded by the \delta_t.

5. While an ablation study regarding the cost parameter 'c' in the Cancer environment is included in Appendix K.8, the process of selecting an appropriate 'c' in a real-world context remains unclear. Practical guidance or heuristic for choosing 'c' would greatly increase the method's applicability and ease of use.

6. The uncertainty threshold \tau is determined from a single episode and subsequently fixed.
It seems that further adjusting \tau as we collect more episodes could potentially enhance the method’s performance. Can the authors comment on this?

7. The submission of code via an external link is not ideal, particularly because the linked material appears to have been last updated on June 11, which is after the supplementary materials deadline.

8. Typos:
- Line 60: “our initial method capable of irregularly observing”, missing “is”
- Appendix 939: “Random that samples the state” should this be “action” instead?
- There seems to be an inconsistency in the numbers:
Line 762: G=1k but G=10k in the main paper



**Questions:**

- See weakness section
- How to choose the parameter \delta_a and \delta_t? Grid search?
- In adaptive quadrature, the time interval for estimating the integral of a function is adjusted based on a comparison of a threshold with the relative estimation error of two different time intervals. Although different to the proposed method, it appears to bear some similarities. Have the authors explored the possibility of applying algorithms from adaptive quadrature to determine the time interval in their proposed method?


**Limitations:**

It is discussed in the abstract and conclusion.

---

> ### Author Rebuttal · Authors · 2023-08-07
>
> Thank you for your thoughtful comments and suggestions!
> ## (A) Dependence on the accuracy of the learned model
> We agree; allow us to clarify that it is standard in all offline model-based RL methods to learn an accurate dynamics model [Levine et al. 2020, Lutter et al., 2021].
>
> We verified empirically that AOC can perform well even when the dynamics model is *less accurate* by performing an *additional experiment ablation* (R1); see the global response. We include this in the supplementary rebuttal pdf. Table 21 shows that AOC with a less accurate dynamics model outperforms Continuous Planning and Discrete Monitoring.
> ## (B) Extending to work with a learned reward model
> You are correct; it is useful to also work in the more general case of learning a reward model. We performed an additional experiment, where we learned a separate reward model from the reward values from the offline dataset—**(R2)**; see the global response. We include this in the supplementary rebuttal pdf. Table 22 shows that AOC with a learned reward model still outperforms Continuous Planning and Discrete Monitoring.
> ## (C) Include related work of Huang et al. 2020, Zhang et al. 2022
> We agree they are relevant and now include them in the paper. In the following (D) we discuss Huang et al. 2020. Zhang et al. 2022 is a discrete planning method that works for *linear* (LQR) systems and optimizes the observing frequency for a given total budget of observations.
> ## (D) Formulation clarification between related work of Huang et al. 2020
> We find the formulation differences between ours and the neat work of Huang et al. 2020 that address an infinite horizon discrete time Linear-Quadratic-Gaussian control problem with observation costs be:
> * Huang et al. 2020 only applies to *linear systems* that are *discrete in time* and have an *infinite time horizon*; our formulation applies to *non-linear systems* that are *continuous in time* and have a *fixed time horizon*.
> * Huang et al. 2020 assume their full system dynamics model is *known*; ours makes no such assumptions and only assumes access to an offline collected dataset of (possibly irregular in time) state-action trajectories to learn a dynamics model, which is more applicable to real-world environments.
>
> We now include a version of this discussion in the related work Section 3.
> ## (E) Clarification of the $\delta_t$ parameter for a continuous time method
> We agree \delta_t can be better explained—it is the continuous search (root finding algorithm) tolerance that is used when searching with binary search for the continuous time duration that the computed action trajectory can be followed for, which occurs when the standard deviation of the reward crosses the threshold $\tau$. We note that all numerical root-finding algorithms generally involve a search tolerance or a similar stopping criterion. The $\delta_t$ tolerance ensures that the binary search algorithm stops evaluations once a solution time is found that is "close enough" to the actual true value. Another way to view the search tolerance $\delta_t$, is that the numerical precision that the search value is correct up to. We included a form of this discussion when introducing $\delta_t$.
> ## (F) Practical guidance for selecting $c$ in a real-world context
> We agree such guidance increases greatly the applicability of the method. The observation cost $c$ should be decided based on the real-world application at hand and include any human preferences in that application if applicable. In real-world systems (with resource constraints), this cost might correspond to actual monetary cost, computational expense, or energy consumption. Further, if this cost involves a human, for example, a patient receiving medical treatment, they may have preferences for treatment, their health impact (e.g., chemotherapy side effects), and or the number, frequency, and timing of treatments that could be included in the cost $c$. Also, there can often be a trade-off between control performance and the number of observations taken, and one way to trade this off is to tune $c$. We have expanded this discussion to create an additional detailed **new Appendix L**, labeled “Practical guidelines to select $c$”.
> ## (G) Updating $\tau$ when more data is collected
> We agree. Currently, we assume an initial offline dataset of state-action trajectories for training the dynamics model. However, we can easily extend this to when more state-action trajectories are collected; we retrain the dynamics model and determine $\tau$ again. This would be most helpful when the initial offline dataset is small and has limited state-action space coverage. We now include this discussion in section 4.
> ## (H) Code submission
> We kindly highlight code submission is *not* compulsory and that the NeurIps policy for code provision is after acceptance, not before. We only updated the readme.
> ## (I) Typos
> Thank you, we agree with all the typos. Yes, line 762 should read $G=10,000$.
> ## (K) How to choose $\delta_a$ and $\delta_t$
> This depends on the real-world environment; $\delta_a$ is often determined for us by the offline dataset of irregular time state-action samples, where it is the mean time between the samples in the dataset. However, if it had to be chosen, it would be a trade-off between the min and max range, where each of the max and min (see Appendix K.4, for decreasing $\delta_a$) lead to poor performance. For the search tolerance $\delta_t$, a sufficiently small value achieves good control performance. Bayesian optimization or grid search would be suitable.
> ## (L) Applying adaptive quadrature algorithms
> We agree; however, our initial method focuses on a simple initial method to verify the theoretical contribution empirically.
> It is precisely more purpose-designed methods we hope to inspire for future work such as this.
> ## Additional References
> * Levine, Sergey, et al. "Offline reinforcement learning" arXiv preprint arXiv:2005.01643 (2020).

---

> > ### Comment · Reviewer_mUFv · 2023-08-17
> >
> > I thank the authors for the comprehensive response, particularly the additional experiments. My concerns have been addressed and I have increased the score accordingly. Regarding the code (readme as explained by the authors), while I don’t think that being non-compulsory grants permission to modify post submission deadline, that is a minor point and did not influence my scoring.

---

> > > ### Author Response · Authors · 2023-08-17
> > > **Gratitude for Revised Review and Score Increase**
> > >
> > > Thank you very much for your thoughtful consideration and the time you have dedicated to reviewing our manuscript. We truly appreciate your recognition of our efforts to address your insightful points, particularly regarding the additional experiments. Your feedback was instrumental in enhancing our work, and we are grateful for your increased score. Thank you once again!

---

### Official Review · Reviewer_KW9G · 2023-07-06

**Soundness:** 3 good
**Presentation:** 3 good
**Contribution:** 2 fair
**Rating:** 6
**Confidence:** 4

**Summary:**

The paper addresses the problem of continuous control with costly observations. The authors provide a formal definition of the observation problem and the control problem. The continue by proposing a scheme to how to do the two simultaneously in irregular intervals. Finally, the method is evaluated against benchmark methods on 4 problems and the results are analyzed. The paper is well structured and clear.

**Strengths:**

The paper deals with an important problem and provide a formal clean definition for it, and also proposes a solution. The experiments not only show that the method has merit but also help to discuss and analyze the solution and the importance of the problem.

**Weaknesses:**

The paper is not clearly stationed within previous work on similar topics. The area of event-based sampling in the control community has delt with close problem and analyzed how to control them and even analyzed the stability of such problem under irregular sampling.

Also, the method in the paper assumes access to the full state (noisy state), which can be unrealistic in real systems (for the purpose of developing theory it is ok but should be discussed). The method assumes the system can be identified and then a controller controls the system for the given interval based on the identified dynamics. This sounds very close to classical control principle such as certainty equivalence and and the separation principle. Those are applied to nonlinear systems as well (while some of the proofs are for linear systems), and since there is no proof of stability or optimality in this paper should be discussed in light of that.

It is unclear what are the limitation of the method in regard of system dynamics, should the system be initially stable? should it be of specific structure?
While the experiments are compared to simple control methods, there are other approaches out there, that addresses similar problem and can be adjusted to this setting. As paper does not compare to other methods (other than naïve ones) it is hard to assess the true value of this work.


**Questions:**

1.	What are the limitation on the dynamics of the system? Can any f(t) be used and the method would work?
2.	The model under non-uniform samples is reduce to semi-MDP, is there something from that line of research that can solve the problem?

---

> ### Author Rebuttal · Authors · 2023-08-07
>
> Thank you for your thoughtful comments and suggestions!
> ## (A) Add similar related work of event-based sampling
> We agree that it is helpful for the reader to expand the related work section to include a discussion of the similar topic of event-based sampling. We now extend our related work section to include the following.
>
> There exists a similar related work area, that of *event-based sampling* in the control community [Åström et al. 1999, Åström et al. 2008], which addresses the similar problem of controlling a system, of only taking a full state observation when an external input *event* occurs, and then providing a control action. To create this *event*, it **assumes part of the state is continuously observed**, and often an *event* is defined when this part of the state (or a function of it) crosses a fixed threshold (e.g., a physical sensor crossing a threshold) [Vasyutynskyy et al. 2010]. This finds multiple uses in problems such as electronic analog control systems for audio and mobile telephone systems [Åström et al. 1999], and battery capacity optimization in wireless devices [Vasyutynskyy et al. 2010]. However, this is different from our proposed problem of continuous-time control whilst deciding when to take costly observations as:
> 1. Event-based sampling *assumes part of the state is continuously observed*. Often in our environments, it is not feasible to observe part of the state continuously (e.g., imaging a cancer tumor volume or performing a medical test) or it is prohibitively expensive to continually observe at a high-frequency part of the state (similar to Continuous Planning approaches).
> 2. The *event* input (the time to take the next observation) is given as a control input to the agent. However, it is precisely the more difficult problem we tackle of coming up with an observing policy that decides *when* to take the next observation.
> 3. The *event* is often defined by a human in advance and is a function of the current partial state. General environments may not have partial state spaces that are predictive of when to observe next, such as a robotic navigation task in two dimensions and only continuously observing one dimension.
> This is now included in Section 3.
> ## (B) Assumption of full noisy state access
> We acknowledge your concern about the assumption of full state access, which may not always be feasible in real-world systems, however is often taken as a standard assumption in other works [Yildiz et al. 2021]. Indeed, we made this assumption primarily to facilitate the development of the theory. We agree, that our proposed problem and formalism could be extended to partial state observations, however, we leave this for exciting future work and development of such methods. We have now included this discussion and assumption in the limitations paragraph in the Conclusion and Future work, section 6.
> ## (C) Proof of Stability or Optimality
> We agree; however, optimality or stability proofs are *not* typically provided for deep RL methods. Allow us to clarify that this current paper contributes the first step to formalizing and understanding theoretically a key property an optimal method should have, which is verified empirically with an initial method.
>
> Furthermore, all model-based RL methods, and RL methods in general, are not optimal in *all scenarios*. A few methods can theoretically find an optimal policy *only in restrictive scenarios*, for example, Q-learning with an MDP, finite and discrete state and action spaces, stationarity, infinite exploration, and a fully observable environment.
>
> We have now added this additional promising future work of a proof of stability and clearly state this as a limitation in the limitation paragraph in the Conclusion and Future work section (section 6).
> ## (D) Limitations of the dynamics of the system
> Our formulation does not impose any specific structure on the dynamics of the system $f$ or require the system to be initially stable. We allow non-linear dynamics and unstable systems. However, we do assume stationarity in the dynamics, i.e., the dynamics are not changing within a single collected offline dataset. Allow us to reiterate that $f$ is defined as a differential equation, which is continuous in time, meaning that existing discrete-time and discrete system dynamics methods are not applicable. We agree this is beneficial to clarify, and we have now added this discussion to the Problem section (section 2).
> ## (E) Related work clarification of Semi-MDP literature
> We agree that semi-MDP literature is a similar related field, where it extends the MDP problem to include *options* that define temporally abstracted action sequences [Sutton et al. 1998]. However, some distinct differences prevent using a semi-MDP to address our continuous-time control whilst deciding when to take costly observations problem. These differences include:
> 1. Semi-MDP is still an underlying discrete MDP with discrete state transitions, whereas our problem formulation focuses on continuous-time systems that can handle continuous actions and states.
> 2. Semi-MDP formulation does not involve an observation cost, whereas our problem formulation does.
> We have now expanded the related work section (section 3) to include the above discussion.
> ## Additional References
> * Åström, Karl Johan, and Bo Bernhardsson. "Comparison of periodic and event based sampling for first-order stochastic systems." IFAC Proceedings Volumes 32.2 (1999): 5006-5011.
> * Aström, Karl J. "Event based control." Analysis and design of nonlinear control systems: In honor of Alberto Isidori. Berlin, Heidelberg: Springer Berlin Heidelberg, 2008. 127-147.
> * Vasyutynskyy, Volodymyr, and Klaus Kabitzsch. "Event-based control: Overview and generic model." 2010 IEEE International Workshop on Factory Communication Systems Proceedings. IEEE, 2010.
> * Sutton, Richard S. "Between MDPs and Semi-MDPs: Learning, planning, and representing knowledge at multiple temporal scales." (1998).

---

> > ### Comment · Reviewer_KW9G · 2023-08-10
> > **Thank you for the response**
> >
> > I would like to thank the authors for the response. No further issues.

---

> > > ### Author Response · Authors · 2023-08-11
> > > **Appreciation for Feedback**
> > >
> > > Thank you very much for your valuable feedback and for taking the time to review our work. We're pleased to hear that you have no further concerns. We hope that our response has successfully addressed all of your questions, and we are optimistic about the impact of our work in the field. We look forward to the final decision and appreciate your consideration. Thank you once again.

---

### Official Review · Reviewer_wAUT · 2023-07-07

**Soundness:** 2 fair
**Presentation:** 2 fair
**Contribution:** 2 fair
**Rating:** 4
**Confidence:** 3

**Summary:**

The paper addresses the problem of controlling continuous-time environments while actively deciding when to take costly observations in time, which is relevant to real-world scenarios such as medicine, low-power systems, and resource management. Existing approaches either rely on continuous-time control methods that take regular, expensive observations in time or discrete-time control with costly observation methods, which are inapplicable to continuous-time settings due to the compounding discretization errors introduced by time discretization. The paper formalizes the continuous-time control problem with costly observations and shows that observing at regular time intervals is not optimal in certain environments, while irregular observation policies yield higher expected utility. The paper proposes an initial method called Active Observing Control (AOC) to solve the problem of continuous-time control with costly observations and demonstrates how AOC can avoid discretization errors in time and achieve a better utility as a result. The paper empirically verifies the key theoretical result in a cancer simulation and standard continuous-time control environments with costly observations.

**Strengths:**

1) The paper addresses an important and unexplored problem of controlling continuous-time environments while actively deciding when to take costly observations in time, which is relevant to real-world scenarios such as medicine, low-power systems, and resource management.
2) The paper provides a theoretical framework for the problem and shows that irregular observation policies can achieve higher expected utility than regular observation policies in certain environments.
3) The paper proposes an initial method called Active Observing Control (AOC) to solve the problem of continuous-time control with costly observations, which can plan action trajectories in continuous time and plan when to observe next in continuous time.
4) The paper demonstrates how AOC can avoid discretization errors in time and achieve a better utility as a result.
5) The paper empirically verifies the key theoretical result in a cancer simulation and standard continuous-time control environments with costly observations.

**Weaknesses:**

1) The paper proposes an initial method called Active Observing Control (AOC) to solve the problem of continuous-time control with costly observations, but determining the optimal method remains an open problem.
2) The paper constructs a simple initial method to solve the problem, with a heuristic threshold on the variance of reward rollouts in an offline continuous-time model-based model predictive control (MPC) planner, which may not be optimal in all scenarios.
3) The paper assumes that the dynamics model is learned accurately from an offline dataset, which may not always be feasible or accurate in practice.
4) The paper only empirically validates the key theoretical result in a cancer simulation and standard continuous-time control environments with costly observations, and further empirical validation in other real-world scenarios is needed to establish the generalizability of the proposed method.

**Questions:**

1) The paper proposes an initial method called Active Observing Control (AOC) to solve the problem of continuous-time control with costly observations, but determining the optimal method remains an open problem.
2) The paper constructs a simple initial method to solve the problem, with a heuristic threshold on the variance of reward rollouts in an offline continuous-time model-based model predictive control (MPC) planner, which may not be optimal in all scenarios.
3) The paper assumes that the dynamics model is learned accurately from an offline dataset, which may not always be feasible or accurate in practice.
4) The paper only empirically validates the key theoretical result in a cancer simulation and standard continuous-time control environments with costly observations, and further empirical validation in other real-world scenarios is needed to establish the generalizability of the proposed method.

**Limitations:**

1) The paper proposes an initial method called Active Observing Control (AOC) to solve the problem of continuous-time control with costly observations, but determining the optimal method remains an open problem.
2) The paper constructs a simple initial method to solve the problem, with a heuristic threshold on the variance of reward rollouts in an offline continuous-time model-based model predictive control (MPC) planner, which may not be optimal in all scenarios.
3) The paper assumes that the dynamics model is learned accurately from an offline dataset, which may not always be feasible or accurate in practice.
4) The paper only empirically validates the key theoretical result in a cancer simulation and standard continuous-time control environments with costly observations, and further empirical validation in other real-world scenarios is needed to establish the generalizability of the proposed method.

---

> ### Author Rebuttal · Authors · 2023-08-05
>
> Thank you for your thoughtful comments and suggestions!
> ## (A) The optimal method remains open
> We agree—in fact, this is the primary motivation for our paper, to first formalize this problem, theoretically contribute a property that an optimal method should have and lay the groundwork for the development of further methods to tackle this “important” and “unexplored” problem of continuous-time control whilst deciding when to take costly observations.
>
> The theoretical contribution of this work is a proof that shows regular observing is not optimal and that irregularly observing can achieve a higher expected utility. We empirically verify this key theoretical result in a cancer simulation and standard continuous-time control environments with costly observations by constructing the simplest initial method to do so (AOC).
>
> Coming up with an optimal method for all scenarios and systems could be *intractable*. Krueger et al. 2020, who focus on a simplified setting of multi-armed bandits where they must pay a cost to observe, propose that for their problem **an optimal method is intractable**—which applies to our problem.
>
> As we agree that the optimal method remains open, we clearly state this throughout the paper, in the *abstract (line 18)*, in the *contributions (line 58)* and in the *conclusion and future work section (line 391)*. We believe solving the optimal problem is out of scope for the current paper (and is most likely intractable, i.e., there is no efficient algorithm to solve it), however, we have provided a key theoretical property and empirically verified, that an optimal method should have.
> ## (B) The simple initial method to solve the problem may not be optimal in all scenarios
> We agree—building on the above rebuttal response **(A)**, such an optimal method for all scenarios may be intractable. Hence, this motivates the need to rely on heuristics to solve the problem, as suggested by Krueger et al. 2020—which is how our initial method (AOC) works, using a heuristic threshold on the variance of the reward rollouts in an offline continuous-time model-based predictive control planner.
>
> Allow us to kindly re-iterate that we explicitly state in the limitations paragraph in line 392 that this “initial solution in our experiments may not be optimal for all scenarios”. Let us restate the official 2023 NeurIps reviewer guidelines, *“In general, authors should be rewarded rather than punished for being up front about the limitations of their work”*. Furthermore, all model-based RL methods, and RL methods in general, are not optimal in *all scenarios*. A few methods can theoretically find an optimal policy *only in restrictive scenarios* (certain conditions), for example, Q-learning with an MDP, finite and discrete state and action spaces, stationarity, infinite exploration, and a fully observable environment [Sutton et al. 2018].
> ## (C) Assumes that the dynamics model is learned accurately from an offline dataset, which may not be feasible or accurate in practice
> We agree and explicitly state this in the limitations paragraph on line 394. This is a key assumption of *all model-based offline RL methods* in general.
> 1.	All offline methods assume that there exists a previously collected dataset of state-action trajectories, as they cannot interact online by definition [Ernst et al. 2005, Levine et al. 2020]. We clearly state throughout the paper (abstract, introduction, contributions, problem definition, etc.) that we are working in the offline setup, where it *is feasible* to have access to an offline dataset. Furthermore, we provide an additional discussion in **Appendix D** of the benefits of offline RL and model-based RL.
> 2.	All offline model-based RL methods rely on the offline dataset covering sufficient state-action space and that the learned dynamics model is accurate enough [Sutton et al. 2018, Levine et al. 2020, Lutter et al., 2021].
>
> We verified empirically that AOC can perform well even when the dynamics model is *less accurate* by performing an *additional experiment ablation* (R1); see the global response. We include this in the supplementary rebuttal pdf. Table 21 shows that AOC with a less accurate dynamics model outperforms Continuous Planning and Discrete Monitoring.
>
> ## (D) Only empirically validates the key theoretical result in a cancer simulation and standard continuous-time control environments with costly observations
> Let us clarify that we empirically validate in more environments than is standard for continuous-time control environments [Yildiz et al. 2021]; as we validate in all three of the continuous-time control environments of the ODE-RL suite from Yildiz et al. 2021, and an additional real-world Cancer environment. Furthermore, we also validate in another environment setup in **Appendix K.3**.
>
> We highlight that we discuss the selection reasons for these environments in **Appendix H**. We selected the standard continuous-time control environments from the ODE-RL suite [Yildiz et al. 2021], which consists of three well-known environments: Pendulum, Cartpole, and Acrobot. Additionally, we implemented a Cancer environment that simulates a Pharmacokinetic-Pharmacodynamic (PK-PD) model of lung cancer tumor growth [Geng et al. 2017] under continuous dosing treatment of chemotherapy and radiotherapy.
>
> As we need a continuous-time environment, many standard discrete-time environments are not applicable. Therefore, we empirically verify all the environments from the ODE-RL suite and an additional cancer environment.
>
> ## Additional References
> * Sutton, Richard S., and Andrew G. Barto. Reinforcement learning: An introduction. MIT press, 2018.
> * Ernst, D., Geurts, P., and Wehenkel, L. (2005). Tree-based batch mode reinforcement learning. Journal of Machine Learning Research, 6(Apr):503–556.
> * Levine, Sergey, et al. "Offline reinforcement learning: Tutorial, review, and perspectives on open problems." arXiv preprint arXiv:2005.01643 (2020).

---

> > ### Author Response · Authors · 2023-08-18
> > **Expanding Empirical Verification: Additional Real-World Validation of Active Observing Control Method in Glucose, HIV, and Quadrotor Environments from Medical and Engineering Domains**
> >
> > Thank you once again for your invaluable insights and time dedicated to the review process. We are thrilled to present **new "further empirical validation in other real-world scenarios"** to address your point 4. Precisely, we have further empirically verified AOC can successfully operate in the real-world environments (medical and engineering) of an accurate **Glucose environment** (controlling the injected insulin for a diabetic patient [Eisen et al. 1988]), a (Human Immunodeficiency Virus) **HIV environment** (controlling the chemotherapy dose for affecting the infectivity of HIV in a patient [Butler et al. 1997]), and a **Quadrotor environment** (controlling the actuators of an unmanned aerial vehicle [Nonami et al. 2010])—this is detailed in response **(E)** below. These new results directly address your concerns and reinforce the generalizability of our AOC method.
> >
> > We have carefully considered all your comments and have worked diligently to address each one. Should any remaining questions or concerns, we welcome the opportunity to clarify them before the author discussion period concludes. Here's a detailed mapping of our responses to your questions:
> > * The optimal method remains open; see **Response (A)**
> > * The simple initial method to solve the problem may not be optimal in all scenarios; see **Response (B)**
> > * Assumes that the dynamics model is learned accurately from an offline dataset, which may not be feasible or accurate in practice; see **Response (C)**
> > * Only empirically validates the key theoretical result in a cancer simulation and standard continuous-time control environments with costly observations; see **Response (D) & (E)**
> >
> > In alignment with your feedback, we have added additional discussions and new results to the paper.
> >
> > We are excited to introduce a **new Appendix O**, entitled **"Further empirical validation in other real-world scenarios"**, to showcase that AOC further empirically validates our main theoretical claim in other real-world environments. We have provided this additional rebuttal point **(E)** below.
> > ## (E) Further empirical validation in other real-world scenarios
> > We specifically chose these three additional environments to respond to your call for further empirical validation in diverse real-world scenarios. Each environment represents a unique challenge and has direct implications in medical and engineering applications, thus reflecting the broad applicability of our method. These are:
> > 1. **Glucose environment**. Controlling the injected insulin for a diabetic patient to regulate their blood glucose level—here observations are costly as a blood test must be performed to measure the glucose level [Eisen et al. 1988].
> > 2. **HIV environment**. Controlling the chemotherapy dose for affecting the infectivity of HIV in a patient—where observations are costly as a blood test must be performed to measure CD4*T cell levels [Butler et al. 1997].
> > 3. **Quadrotor environment**. Controlling the actuators of an unmanned aerial vehicle—where observations can be costly due to performing an expensive (power and compute) localization measure [Nonami et al. 2010].
> >
> > We observe that AOC still achieves a high average utility $\mathcal{U}$ on all environments, outperforming the competing Continuous Planning and Discrete Monitoring methods, reinforcing our theoretical claims, and extending our empirical validation.
> >
> > **Table 24. Glucose Environment**
> > |Policy|$\mathcal{U}$|$\mathcal{R}$|$\mathcal{O}$|
> > |-|-|-|-|
> > |Random|0$\pm$0|0$\pm$0|50$\pm$0|
> > |Discrete Planning|96$\pm$0.485|96$\pm$0.485|50$\pm$0|
> > |Discrete Monitoring|120$\pm$0.489|92.9$\pm$0.493|15.7$\pm$0.0466|
> > |Continuous Planning|100$\pm$0.39|100$\pm$0.39|50$\pm$0|
> > |**Active Observing Control**|**126$\pm$0.41**|99.9$\pm$0.39|17.5$\pm$0.0336|
> >
> > **Table 25. HIV Environment**
> > |Policy|$\mathcal{U}$|$\mathcal{R}$|$\mathcal{O}$|
> > |-|-|-|-|
> > |Random|0$\pm$0|0$\pm$0|50$\pm$0|
> > |Discrete Planning|152$\pm$0.121|152$\pm$0.121|50$\pm$0|
> > |Discrete Monitoring|2.77e+03$\pm$0.455|126$\pm$0.455|7$\pm$0|
> > |Continuous Planning|100$\pm$0.431|100$\pm$0.431|50$\pm$0|
> > |**Active Observing Control**|**2.83e+03$\pm$1.56**|107$\pm$0.878|5.75$\pm$0.0269|
> >
> > **Table 26. Quadrotor Environment**
> > |Policy|$\mathcal{U}$|$\mathcal{R}$|$\mathcal{O}$|
> > |-|-|-|-|
> > |Random|0$\pm$0|0$\pm$0|50$\pm$0|
> > |Discrete Planning|101$\pm$0.00882|101$\pm$0.00882|50$\pm$0|
> > |Discrete Monitoring|1.79e+03$\pm$0.199|101$\pm$0.0142|5.99$\pm$0.00518|
> > |Continuous Planning|100$\pm$0.015|100$\pm$0.015|50$\pm$0|
> > |**Active Observing Control**|**1.83e+03$\pm$0.0789**|97.9$\pm$0.0258|5$\pm$0.00196|
> >
> > We believe that these new experimental results, conducted in alignment with your feedback, not only bolster our empirical validation but also emphasize our method's practical utility and theoretical integrity. We hope this additional evidence adequately addresses your concerns and merits reconsidering the initial score. Should any uncertainties linger, we remain at your disposal for further clarification; thank you!

---

> > ### Comment · Reviewer_wAUT · 2023-08-21
> > **Response to author**
> >
> > Thanks for your explanations. I would like to keep my original score

---

### Official Review · Reviewer_U2ft · 2023-07-10

**Soundness:** 2 fair
**Presentation:** 3 good
**Contribution:** 2 fair
**Rating:** 6
**Confidence:** 3

**Summary:**

The authors present a continuous-time framework for "active observation", meaning deciding when to take observations in a control problem, assuming taking observations has a cost. They present two separated controllers, one for taking action and one for deciding when to observe. The controller for acting is an MPC controller, the controller for deciding when to observe is based on binary search. They learn a continuous-time stochastic dynamics model using an MLP with delta time in the input, with ensembling to model epistemic uncertainty and a gaussian prediction to model aleatoric uncertainty.

**Strengths:**

The problem is really interesting, and it's obvious from the introduction that this is something that should be worked on. The paper is well-written.

**Weaknesses:**

Let's start with a few things that I think are incorrect in the math. Line 203 seems to state that a mixture of gaussian (summing the pdfs, with uniform weight) is a gaussian. That is not correct. You can compute the mean and variance of the mixture, as your equations suggest, but that doesn't make it a gaussian.

Same thinking from 241 to 246. With your ensembling, the z distributions at t^{k+1} won't be gaussian, and neither will be your reward. Yes you can compute their mean and covariance, but the reward distribution won't be gaussian unless your state distribution is gaussian and the reward function is linear.

Beyond the dynamics model which is truly continuous-time (can predict future observation at an arbitrary floating point resolution), I think the proposed control algorithm is decisively discrete-time. The delta_t for observing is smaller than delta_a for control, but that's it: these are two discrete-time controller with differently size timesteps. There is no notion of adaptive step size during control, the controller makes no use of the continuous time formulation in either case. The advantage for the approach, if I understand correctly, does not come from the continuous-time formulation, but merely for allowing the observation controller to control at a higher frequency than the action controller.

The dynamics model is only conditioned on the previous observation z. But in general, you need the whole history of noisy observations and action to predict the next, especially in noisy scenarios.

The formulation is limited by the fact that the observation is simply the true state plus some gaussian noise, rather than being a non-linear transformation of the state.

The claim that ensembling captures epistemic and gaussian prediction captures aleatoric uncertainty is vague to me. Intuitively that both should capture both type of uncertainty to some degree; can you provide a reference?

Table 3: the bold font should be applied to continuous planning when it performs better or just as well. So continuous planning - cancer reward, acrobot reward, and pendulum reward should be bold.

In Krueger et al, they actively sample rewards, not observations

Line 73, the wiggle should be an equal sign



**Questions:**

For the experiment in figure 3, what happens if you use discrete monitoring with a slightly smaller timestep? Same for figure 4; to me this feels simply like the time discretization is too big and the gain is simply from the fact that the observation controller has a higher frequency.

How would you handle the interesting case where the cost of observing depends on the state?

How would you create a combined act+observe controller?

**Limitations:**

The authors are honest about the limitations of their method. I would add the limitation of the observation simply being the true state plus some noise.

---

> ### Author Rebuttal · Authors · 2023-08-04
>
> Thank you for your thoughtful comments and suggestions!
> ## (A) Distributions of state and reward
> We agree that a uniformly weighted mixture of Gaussians is not a Gaussian itself, rather it is possible to compute the mean and variance of the mixture to approximate it with a Gaussian, and we have now fixed this typo in line 203.
>
> Allow us first to clarify that we follow the standard model-based uncertainty-aware dynamics model methodological setup from the seminal work of Chau et al. 2018. By doing so, we perform Monte Carlo Sampling of simulating state particle $z_p(t^{(k+1)})$, roll-outs of state trajectories, *to capture the multi-modality* of the state and reward distributions [Chau et al. 2018]. You are correct that when forward propagating to determine the future state and reward distributions, the distribution of the state particles $z_p(t^{(k+1)})$ is not Gaussian. Although we do compute the standard deviation of the reward particles at an evaluation time, we find this *approximate* statistic to empirically work well in AOC—which is the *simplest* instantiation of a method to solve this new problem setting and verify the key theoretical contribution that irregular observing policies can achieve a higher utility than regular observing policies.
>
> ## (B) Why can’t you create a discrete observing policy at a higher frequency?
> We agree that creating a discrete observing policy with a higher frequency is possible, and we already provide an experiment in **Appendix K.4** to show this. Allow us to re-iterate the goal of the paper is to formalize the problem of continuous-time control with observation costs, provide the theoretical proof that irregular observing policies can achieve a higher utility than regularly observing policies, and verify this key theoretical contribution with the simplest initial instantiation of a method, that of AOC.
>
> The inherent benefit of the continuous-time dynamics model is that it allows arbitrary evaluation at any future time, which is beneficial when determining the time that the reward variance crosses the threshold, which is determined up to a tolerance of $\delta_t$, as is common for any root finding algorithms (e.g., Newton’s Method).
>
> However, using an observing policy that is discrete at a higher frequency is **not practical**, as $\delta_t << \delta_a$, computing the reward variance at every $\delta_t$ within the whole action trajectory duration $H$ scales poorly—commonly known as *linear search*, and scales as $\mathcal{O}(n)$, where n is the worst case number of evaluations of the reward variance, $n = \lfloor \frac{H}{\delta_t} \rfloor$. Instead, AOC employs the simplest adaptive search method, that of *binary search*, which scales as $\mathcal{O}(\log(n))$, which **is practical**. We now include this discussion in Section 4.2.
> ## (C) Discrete monitoring with a smaller timestep
> Building on the above, we already provide empirical results in **Appendix K.4** (line 1036) for discrete monitoring with a smaller timestep $\delta_a$. As we reduce $\delta_a$, all baselines degrade in performance; however, AOC still has the highest utility amongst the baselines for a set $\delta_a$—with the gain over discrete monitoring reducing. The improvement gain of AOC compared to discrete monitoring arises from:
> 1. Avoiding time discretization errors, finding $\rho(z(t_i))^*$ up to a tolerance of $\delta_t$ compared to a large discretization of $\delta_a$ (where $\delta_t << \delta_a$)
> 2. scales much better as the tolerance $\delta_t$ is reduced, $\mathcal{O}(log(n))$ versus discrete monitoring of $\mathcal{O}(log(n))$
> 3. the continuous-time dynamics model is more accurate as it can learn from the irregularly observed offline dataset, whereas the discrete-time dynamics model does not use $\delta$ as an input.
> ## (D) Dynamics model is only conditioned on the previous observation
> The implementation dynamics model choice does not affect how the different observing policies $\rho$ compare, as they *all use the same form of dynamics model*. We chose this as the simplest dynamics model, which happens to be a Markovian neural network. However, we could have used an RNN-based continuous-time model to encode the whole history of observations.
> ## (E) Non-linear state transformations
> Let us clarify that we follow the standard assumptions for an environment description in control [Yildiz et al., 2021]. We agree that observations cannot be non-linear in the current formulation. However, it *does allow for non-linear state transitions*. We have clarified this in Section 2.
> ## (F) Capturing both aleatoric and epistemic uncertainty with the probabilistic dynamics model
> Let us clarify that this is well known and comes from the key reference of Chau et al. 2018, which provides an excellent explanation in their Section 4, from the top of page 4—we reference it on lines 180,188.
> ## (G) Clarification of related work w.r.t. Krueuger at al. 2020
> Krueger et al. 2020, focus on the simpler problem of multi-armed bandits (MAB), where there is a cost to observe the reward. In the MAB setting, it is possible to formulate it as an RL PO-MDP with one state. The underlying static state (the fixed reward distributions of each bandit) is unknown and must be determined by taking successive costly observations (paying a cost to observe a reward). Therefore, Kreuger et al. 2020’s statement that the optimal algorithm for their simple MAB setting is intractable applies to our problem. We now clarify this in Section 3.
> ## (H) Extensions
> Handling state-dependent observation costs requires intricate reward scaling. A combined act+observe controller with MPC would scale poorly in complexity. These are beyond our current scope but are valuable directions for future work. We thank the reviewer for their suggestions.
> ## (I) Typos
> Thank you, we will un-bold the reward in Table 3, as only Utility should be bolded as that is our ultimate objective to maximize; yes, line 73 should be $=$.

---

> > ### Author Response · Authors · 2023-08-16
> > **Empirical Verification of AOC with Non-Linear State Transformations and Responses to Reviewer Comments**
> >
> > Thank you once again for your invaluable insights and time dedicated to the review process. We are thrilled to present a **new** experiment that empirically verifies that AOC can successfully operate in environments where *observations are non-linear transformations of the state*, denoted by response **(J)** below.
> >
> > We have carefully considered all your comments and have worked diligently to address each one. Should there be any remaining questions or concerns, we welcome the opportunity to clarify them before the author discussion period concludes. Here is a detailed mapping of our responses to your questions:
> >
> > * Distributions of state and reward; see **Response (A)**
> > * Creating a discrete observing policy at a higher frequency; see **Response (B)**
> > * Discrete monitoring with a smaller timestep; see **Response (C)**
> > * Conditioning of the dynamics model on the previous observation; see **Response (D)**
> > * Non-linear state transformations; see **Response (E)**
> > * Handling aleatoric and epistemic uncertainty with the probabilistic dynamics model; see **Response (F)**
> > * Clarification regarding related work such as Krueger et al. 2020; see **Response (G)**
> > * Extensions; see **Response (H)**
> > * Typos; see **Response (I)**
> >
> > In alignment with your feedback, we have updated our manuscript with pertinent discussions and clarifications.
> >
> > We are excited to introduce a **new Appendix N**, entitled **"AOC also empirically works for non-linear state transformations"**, to empirically showcase that AOC can indeed be applied in environments characterized by non-linear state transformations. We have provided this additional rebuttal point (J) below.
> >
> > ## (J) AOC also empirically works for non-linear state transformations
> >
> > Our latest experiment investigates AOC's performance within environments that utilize observations stemming from non-linear state transformations. We tailored the existing Cancer environment to render observations via the non-linear state transformation function $z(t)=0.1(s(t) + \epsilon(t))^2 + (s(t) + \epsilon(t))$. The subsequent results, conducted across 1,000 random seeds, are outlined in Table 23 below.  We observe that AOC still achieves a high average utility $\mathcal{U}$ on all environments, outperforming the competing Continuous Planning and Discrete Monitoring methods. Specifically, this empirically further verifies our key theoretical contribution that regular observing is not optimal and that irregularly observing can achieve a higher expected utility.
> >
> > **Table 23**
> >
> > | Policy                   | $\mathcal{U}$ | $\mathcal{R}$ | $\mathcal{O}$ |
> > |--------------------------|---------------|---------------|---------------|
> > | Random                   | 0$\pm$0        | 0$\pm$0        | 13$\pm$0       |
> > | Discrete Planning        | 91.7$\pm$0.397 | 91.7$\pm$0.397 | 13$\pm$0       |
> > | Discrete Monitoring      | 90.5$\pm$0.559 | 85.4$\pm$0.548 | 5.25$\pm$0.0329 |
> > | Continuous Planning      | 100$\pm$0.21   | 100$\pm$0.21   | 13$\pm$0       |
> > | **Active Observing Control** | **104$\pm$0.221**  | **98.9$\pm$0.208** | **4.68$\pm$0.0315** |
> >
> > Should any uncertainties linger, we sincerely invite you to share them with us before the author discussion period concludes in the next few days. Your continued engagement is deeply appreciated, and we are at your disposal for any further elucidation, thank you!

---

> > > ### Comment · Reviewer_U2ft · 2023-08-16
> > > **Score increase**
> > >
> > > Thank you to the authors and I apologize for the very short (and a bit late) response. I believe the authors made several good efforts to address my points and I am increasing my score.

---

> > > > ### Author Response · Authors · 2023-08-17
> > > > **Gratitude for Revised Review and Score Increase**
> > > >
> > > > Thank you very much for your thoughtful consideration and the time you've dedicated to our manuscript. We truly appreciate your recognition of our efforts to address your insightful points. Your feedback was instrumental in enhancing our work, and we are grateful for your increased score by two points. If there are any more questions or comments in the future, please don't hesitate to reach out. Thank you once again!

---

### Author Rebuttal · Authors · 2023-08-07

We thank all reviewers for their thoughtful comments and suggestions! We respond to each reviewer with an individual rebuttal and share all **additional new experimental** results here.

## (R1) Dependence on the accuracy of the learned dynamics model
To understand this further, we performed an *additional experiment ablation*, where we benchmarked all methods with a less accurate dynamics model—training all the dynamics models with fewer samples (10\% of the total amount of samples used in training the dynamics models presented in the main paper), here trained on 100,000 samples.

We provide the experimental results in the rebuttal supplemental pdf as Table 21.
We observe that AOC still achieves a high average utility $\mathcal{U}$ on all environments, outperforming the competing Continuous Planning and Discrete Monitoring methods.
Specifically, this empirically further verifies our key theoretical contribution that regular observing is not optimal and that irregularly observing can achieve a higher expected utility. We now include this discussion and new experiment in an **additional new Appendix L**.

## (R2) Extending AOC to work with a learned reward model
To investigate whether AOC can also be used with a learned reward model, we performed an *additional experiment*, where we trained an MLP reward model (4-layer MLP with 128 units and Tanh activations) from the offline dataset in all the benchmarks.

We provide the experimental results in the rebuttal supplemental pdf as Table 22.
We observe that AOC still achieves a high average utility $\mathcal{U}$ on the Cancer environment, outperforming the competing Continuous Planning and Discrete Monitoring methods.
This empirically verifies that our proposed initial approach can still perform well using a learned reward model and that our theoretical contribution still holds. We now include this additional experiment in an **additional new Appendix M**.

---

### Decision · Program_Chairs · 2023-09-21

**Decision:**

Accept (poster)

**Comment:**

This paper formalizes the problem of continuous-time control with observation costs (for non-linear, continuous-time systems with fixed time horizon and unknown dynamics), proves theoretically that observing at regular time intervals is not always optimal, constructs an initial method (using MPC for selecting actions and a binary search-based method to determine when observations should be taken), and validates this method on four continuous-time control environments (three from ODE-RL and a cancer environment). The reviewers agreed that the statement of this problem is interesting and novel, with solid theoretical and experimental insight. While they raised several issues in their initial reviews -- including dependence on the accuracy of the learned model, uncertainty as to whether the proposed control algorithm is actually discrete-time, questions about the relationship to prior work proposing a similar formulation, and questions about/corrections to several mathematical details -- the authors were able to address these concerns to the reviewers' satisfaction through additional experiments and adding additional discussion/clarification to the paper.